# INCENTIVIZED BLACK-BOX MODEL SHARING

## ABSTRACT

Black-box model sharing is a preferable alternative to data sharing because of practical considerations (e.g., administrative regulation and data expiration). However, previous works may neglect the self-interests of individual parties. To encourage self-interested parties to contribute predictions in the ensemble, it is crucial to provide incentives, such as *fairness*: allocating higher reward/payoff to parties with more contributions, and *individual rationality*: ensuring guaranteed model performance improvement for each party. This paper presents a novel incentivized black-box model sharing framework that fairly distributes ensemble predictions and monetary payoffs commensurate to each party's contribution. We propose a contribution measure using the average ensemble weight of black-box models. Subsequently, we derive a closed-form solution that explicitly determines the fair reward and payoff allocation given the contribution and payment. By incorporating ensemble predictions and analyzing the generalization error bound, we theoretically show approximate individual rationality is guaranteed. Furthermore, we empirically demonstrate our proposed method achieves incentive guarantee using real-world datasets.

## 1 INTRODUCTION

Collaborative machine learning is a promising approach that allows different parties to collaboratively build ML models. Although it is possible for parties to facilitate data sharing (Doel et al., 2017; Tenopir et al., 2020), they may not share or sell their private data due to administrative regulation and/or data expiration (GDPR.EU, 2018). For example, different hospitals cannot share data to optimize patient treatment, if data sharing is prohibited due to confidentiality concerns (Mulligan, 2001) or the training data are expired at the end of clinical studies (Hulsen, 2020). As ML models are built to learn from the training data, model sharing can help address concerns of data sharing and offer a more attractive collaboration method (Li et al., 2022). Nevertheless, parties may also hesitate to disclose their model parameters due to concerns about potential information leakage (Hitaj et al., 2017; Zhu et al., 2019). Therefore, sharing *black-box* models (i.e., the internal parameters are not disclosed, and only the predictions given input data can be observed) is a more appealing choice because it discloses less model information.

Current works on black-box model sharing (Feng et al., 2021; Chang et al., 2021; Lin et al., 2020; Li et al., 2021; Papernot et al., 2017) aggregate (e.g., mean or weighted average) the predictions of individual models into *ensemble predictions*, which are then distributed to all parties as synthetic data to improve their respective model performance. These works rely on an assumption that all parties willingly contribute their predictions, which is unfortunately difficult to satisfy in practice. Consider a hypothetical scenario in which one primary party, such as Alice, contributes her predictions, while the remaining parties contribute nothing (e.g., multiple student models distill one teacher model (Chang et al., 2022; You et al., 2018)). As Alice cannot improve her model from such collaboration, she is not motivated to contribute her predictions, especially since the data collection and training computation of ML models can incur substantial costs. In this regard, self-interested parties require suitable incentives (e.g., an improved model or financial rewards) to collaborate (Lo & DeMets, 2016), motivating the need for an incentive scheme in black-box model sharing.

Drawing parallels between data sharing (Sim et al., 2020) and black-box model sharing, we identify two similar key incentives as *fairness* and *individual rationality* (IR). Fairness suggests that every party should receive a reward proportional to its contribution (Sim et al., 2020). To distribute a fair reward, it is imperative to first determine the contribution of each party to the collaboration. Existing

works of data sharing use data quality as the contribution measure, which requires partial/full access to private data (Ghorbani & Zou, 2019; Yoon et al., 2020) or model-related information, such as Bayesian networks (Sim et al., 2020) and K-nearest neighbor classifiers (Jia et al., 2019b). However, such access to data or information on models is *not* available in black-box model sharing. Then, (1) *how should we measure the contributions in black-box model sharing?*

Furthermore, the type of reward offered in collaboration is a crucial factor in incentivizing collaboration. While monetary payoffs are often used in collaboration (Han et al., 2023; Liu et al., 2021), where less-contributing parties are required to compensate top contributors financially, some parties may lack sufficient budget to cover the costs or prefer improving model performance to acquiring payoffs (Sim et al., 2020). On the other hand, non-monetary reward schemes, such as rewarding by varying sizes of synthetic data (Tay et al., 2022), aim to ensure fairness by proportionally distributing rewards. However, this may restrict parties with adequate monetary resources from leveraging their budgets to obtain additional rewards. For instance, a reward mechanism that incorporates monetary payoffs can extend to model marketplaces (Liu et al., 2021; Agarwal et al., 2019), where it allows non-contributing parties with ample financial budgets to compensate model sellers for their contributed models. Accordingly, jointly considering monetary payoffs and non-monetary rewards can address these restrictions, and enable the parties to acquire what they lack by offering what they have. To satisfy fairness in this joint allocation scheme, a crucial question is: (2) *what is a fair payment for acquiring additional rewards in the collaborative setting?*

Finally, IR suggests that no party will be worse off by collaborating (Sim et al., 2020). In black-box model sharing, it means the model performance of each party will be improved after receiving rewards (in the form of predictions). However, if the ensemble predictions (i.e., rewards) are of low quality (e.g., many inaccurate models are involved in collaboration), model performance might degrade when ensemble predictions are directly used as additional training data by the parties and violate IR. Thus, (3) *how can IR be satisfied in black-box model sharing?*

The address the three questions, we propose a framework of *incentivized black-box model sharing*. For (1), we introduce a *Weighted Ensemble Game* (WEG) using the average ensemble weight to quantify the contribution of black-box models towards the ensemble predictions, which is validated by our study showing the quality of ensemble predictions depends on individual ensemble weights. For (2), we suggest that the sum of the reward and payoff of each party is proportional to its contribution. To ensure a fair ratio between payments and rewards, we then propose *Fair Replication Game* (FRG) to characterize the value of the bought rewards and payoffs. We show that the Shapley value for the combined game of WEG and FRG has a closed-form solution that concisely specifies the allocation of reward and payoff, and satisfies fairness. For (3), we theoretically show $\epsilon$-IR, a relaxed version of IR, is satisfied by analyzing the generalization (error) bound. We also empirically demonstrate the appealing attributes and efficacy of our proposed scheme in collaboration using real-world datasets.

## 2  RELATED WORK

**Black-Box Model Sharing**. In the study of unsupervised domain adaptation from black-box models, numerous works (Feng et al., 2021; Ahmed et al., 2021; Liang et al., 2022) have facilitated black-box model sharing to improve model performance and address privacy concerns, where they transfer knowledge to the unlabeled target domain from black-box models. Concurrently, federated learning with ensemble distillation (Chang et al., 2021; Lin et al., 2020; Li et al., 2021; Li & Wang, 2019) has also attracted considerable attention for its ability to produce a single, distilled model from multiple black-box models in a distributed manner. The literature in this domain has primarily concerned the performance improvement of ensemble distillation while adhering to the restrictions on privacy preservation and communication efficiency. The primary motivation behind these preceding works has been to address practical concerns (e.g., privacy). However, it is essential to recognize that parties might also have self-interested motivations in practice. Our study serves as the first exploration of incentive-aware collaboration within the context of black-box model sharing.

**Valuation Problem**. Various methods (Ghorbani & Zou, 2019; Jia et al., 2019a; Xu et al., 2021; Kwon & Zou, 2022; Wu et al., 2022) have been proposed for data valuation for a variety of tasks. These methods, which have been proven both accurate and beneficial, are particularly relevant in the context of data sharing. Yet, little research has explored the topic of black-box model valuation. To the best of our knowledge, a solitary study (Rozemberczki & Sarkar, 2021) has considered black-box

model valuation within an ensemble game framework, employing weighted voting to quantify the value of multiple binary classifiers. However, our work covers a more general classification problem and theoretically studies how model contributions affect ensemble performance.

**Incentive Mechanism in CML**. Prior work focused on how to distribute only either monetary payoffs (Han et al., 2023; Cai et al., 2015; Jia et al., 2019c; Zhan et al., 2020) or non-monetary rewards (Sim et al., 2020; Tay et al., 2022; Karimireddy et al., 2022) while ensuring some incentive objectives (e.g., fairness and IR). In particular, (Sim et al., 2020; Tay et al., 2022) distribute rewards based on Shapley value to incentivize collaboration. There lacks a unified scheme to fairly bridge monetary payoffs and non-monetary rewards in the collaboration. Towards this goal, (Nguyen et al., 2022) proposed conditional Shaley value to adjust model rewards by using linear programming under budget constraints, but we will show that it did not satisfy Shapley fairness in jointly allocating rewards and payoffs. However, our method theoretically and empirically satisfies the Shapley fairness. We also empirically show we satisfy the fairness incentive compared to this baseline method. Moreover, our method has a closed-form solution that specifies the exact reward and payoff allocation. We also recognize a distantly related study on the model-sharing game (Donahue & Kleinberg, 2021). However, their research framework focuses on sharing white-box models and does not address fairness and IR incentives.

## 3 SETTING, BACKGROUND AND OVERVIEW

We consider a set $N := \{1, ..., n\}$ of $n$ self-interested parties. Each party $i \in N$ has a predictive model $h_i \in \mathcal{H} : \mathcal{X} \mapsto \triangle^{K-1}$ where $\mathcal{H}$ is the hypothesis space, $\mathcal{X}$ is the input space, and $\triangle^{K-1}$ is the $K$-probability simplex. The predictive model $h_i$ is trained to minimize the empirical risk with $m_i$ sample sizes on its source domain $\langle \mathcal{D}_i, f \rangle$ where $\mathcal{D}_i$ is its data distribution and $f$ is the same labeling function across all source domains. We denote by $\langle \mathcal{D}, f \rangle$ the target domain on which *all parties* want to predict well, and $U \sim \mathcal{D}$ a set of *unlabeled* data of size $T$ i.i.d. sampled from $\mathcal{D}$.

Updated: An overview of our proposed 2-stage mechanism is illustrated in Fig. 1: In **stage 1** (in Sec. 4), a trusted host queries all the parties on each $x \in U$ to obtain their corresponding predictions $\{h_i(x)\}_{i=1}^n$, to produce an ensemble prediction $h_N(x) = \sum_{i=1}^n \beta_{i,x} h_i(x)$ where $\beta_{i,x}$ is the ensemble weight of $h_i(x)$ determined by a *given* choice of the ensemble (elaborated later). In this way, each party $i$ makes contributions by providing predictions $h_i(x)$, and we make use of the Shapley value $\phi_i$ (Shapley, 1953) to *fairly* quantify the value of their *contributions* (where the fairness is formalized via certain properties in App. B.1) by designing a suitable valuation function $\mathcal{V}$ (that depends on $h_i(x)$): $\phi_i(\mathcal{V}) := (1/n!) \sum_{\pi \in \Pi_N} [\mathcal{V}(C_{\pi,i} \cup \{i\}) - \mathcal{V}(C_{\pi,i})]$ where $\Pi_N$ is the set of all possible permutations of $N$ and $C_{\pi,i}$ is the coalition of all parties preceding $i$ in the permutation $\pi$. In **stage 2** (in Sec. 5), based on the contribution $\phi_i$ of each party, we design a fair reward (of value) $r_i \in \mathbb{R}_+$ and realize the reward in the form of an i.i.d. subsampled set of the *ensemble predictions* $\{x_t, h_N(x_t)\}_{t=1}^T$. Moreover, we enable each party to make monetary payment $p_i \in \mathbb{R}_+$ (through the trusted host) to other parties to obtain additional rewards (of value) $r_i^+ \in \mathbb{R}_+$. In particular, we show that the reward and payoff allocation satisfies *fairness* and $\epsilon$-*IR* incentives.

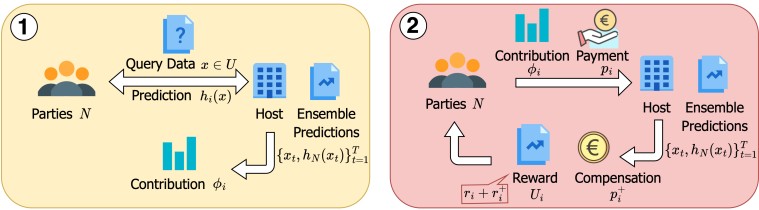

Figure 1: Overview of incentivized black-box model sharing framework. Stage 1 (left) uses the parties' predictions $h_i(x)$'s to obtain the contributions $\phi_i$'s (Sec. 4). Stage 2 (right) uses the contributions $\phi_i$ and payments $p_i$ to realize the reward $U_i$ (of value $r_i + r_i^+$) and compensation $p_i^+$ (Sec. 5).

## 4 CONTRIBUTION EVALUATION IN BLACK-BOX MODEL ENSEMBLE

### 4.1 ENSEMBLE METHOD

We observe that the weighted sum formulation $h_N(x) = \sum_{i=1}^n \beta_{i,x} h_i(x)$ appears in many existing ensemble methods: In average ensemble (AVG) (Lin et al., 2020), $h_N(x) = n^{-1} \sum_{i=1}^n h_i(x)$ where

$\beta_{i,x} = 1/n$. In multiplicative weight update (MWU) (Blum et al., 2021), $\beta_{i,x}^t = -\log(\|h_N^t(x) - h_i(x)\|_2 / \sum_{i=1}^n \|h_N^t(x) - h_i(x)\|_2)$ at each weight update step $t$. More examples are in App. B.2. As different ensemble methods may yield predictions of varying qualities, we introduce here an optimal ensemble method used in our experiments in Sec. 6. When the ground truth of $U$ is known by the host, the optimal ensemble weight $\{\beta_{i,x}^*\}_{i=1}^n$ for $x \in U$ is obtained by solving the following linear optimization problem: minimize $|f(x) - \sum_{i=1}^n \beta_{i,x}^* h_i(x)|$ over $\{\beta_{i,x}^*\}_{i=1}^n \in \triangle^{n-1}$ .

## 4.2 FAIR CONTRIBUTIONS IN ENSEMBLE

We first identify the *additive* structure from ensemble predictions and further exploit the additivity to define a novel class of cooperative games, where parties aggregate their predictions via weighted sum for a given distillation dataset (e.g., $U$).

**Definition 4.1** (Weighted Ensemble Game). *For the ensemble weight $\beta$ specified by certain ensemble methods, the weighted ensemble game for a given dataset $U$ with size $T$ is $G = (N, \mathcal{V}, U, \beta)$ and the valuation function $\mathcal{V}$ is given by*

$$\mathcal{V}(C; U) := \sum_{x \in U} \left\| \sum_{i \in C} \beta_{i,x} h_i(x) \right\|_1 / T = \sum_{x \in U} \sum_{i \in C} \beta_{i,x}/T, \ \forall C \subseteq N .$$

Let us first consider the simplest case when only one data point is given for the game (i.e., $T = 1$). Thus, the $\ell^1$-norm of the weighted prediction $\sum_{i \in C} \beta_{i,x} h_i(x)$ measures the importance/weight of coalition $C$ to the ensemble prediction $h_N(x)$. Note that $\mathcal{V}(C; \{x\})$ is equivalent to $\sum_{i \in C} \beta_{i,x}$ as each $h_i(x)$ is a probability vector and $\beta_{i,x}$ is non-negative. So, if $\mathcal{V}(C; \{x\})$ is large, the coalition $C$ contributes more valuable predictions to $h_N(x)$. In black-box model sharing, the host gathers the predictions $\{h_i(x)\}_{x \in U}$ from each party for the dataset $U$. Therefore, $\mathcal{V}(C; U)$ measures the contribution of coalition $C$ to all ensemble predictions. We let $\mathcal{V}_i = \mathcal{V}(\{i\}; U)$ to ease notation. As a concrete example, $\mathcal{V}_i = \sum_{x \in U} \beta_{i,x}/T$, which is the average ensemble weight of model $h_i$ and represents the contribution of party $i$ to ensemble predictions.

The function $\mathcal{V}$ satisfies the standard assumptions of CGT, i.e. $\mathcal{V}(\emptyset) = 0$ and $\mathcal{V}(C) \geq 0 \ \forall C \subseteq N$, making it compatible with solution concepts from CGT. To *fairly* measure the contribution of each party in the grand coalition $N$, we use the Shapley value recalled in Sec. 3, which is unfortunately computationally expensive to calculate (Kwon & Zou, 2022). By exploiting the additivity of $\mathcal{V}$ (i.e., $\mathcal{V}(C_1 \cup C_2) = \mathcal{V}(C_1) + \mathcal{V}(C_2)$ for every coalition $C_1 \subseteq N$ and $C_2 \subseteq N \setminus C_1$), we can obtain the Shapley value in linear time, and its analytic form for each party $i$ is given as $\phi_i(\mathcal{V}) = \mathcal{V}_i = \sum_{x \in U} \beta_{i,x}/T$. The proof is in App. C.5. We use the shorthand $\phi_i = \phi_i(\mathcal{V})$ only for $\mathcal{V}$ when the context is clear. Therefore, the fair contribution of a black-box model in the ensemble is measured by its average ensemble weight. One might initially think our formulation of $\mathcal{V}$ as simplistic, but its additive property enables us to obtain the closed form of $\phi_i$. A recent work Drungilas et al. (2023) proposed to determine the ensemble weight $\beta_{i,x}$ by measuring the contribution of a learner to an ensemble with the Shapley value. Therefore, our valuation method that directly uses the ensemble weight as the contribution is meaningful. Moreover, the formulation of $\mathcal{V}$ can also be broadly applied to numerous ensemble methods as shown in Table 1, 4, and 5.

## 4.3 GENERALIZATION BOUND FOR WEIGHTED ENSEMBLE

Next, we provide further justification of our model valuation method by analyzing the generalization bound for the virtual ensemble model $h_N$. We first introduce some notations: the error of a hypothesis $h$ on the target domain $\langle \mathcal{D}, f \rangle$ is $L_\mathcal{D}(h, f) := \mathbb{E}_{x \sim \mathcal{D}}[\ell(h(x), f(x))]$ where $\ell$ is the loss function and $f$ is the labeling function. We use the shorthand $L_\mathcal{D}(h) = L_\mathcal{D}(h, f)$ next.

**Proposition 1.** *For the ensemble model $h_N$ and for any $\delta \in (0, 1)$, with probability $\geq 1 - n\delta$:*

$$L_\mathcal{D}(h_N) \leq \sum_{i=1}^n \left( \mathcal{V}_i + \sqrt{2\log(2/\delta)/T} \right) L_\mathcal{D}(h_i) + \Sigma_N$$

*where $\Sigma_N := \sum_{i=1}^n \mathbb{E}_{x \sim \mathcal{D}}[(\beta_{i,x} - \mathbb{E}_\mathcal{D}[\beta_{i,x}])(|h_i(x) - f(x)| - L_\mathcal{D}(h_i))]$ .*

The proof is given in App. C.1. The constant $\delta$ is due to the probability over the choice of samples, and $\Sigma_N$ is the sum of some covariances. The (upper bound of) generalization error of the ensemble model $h_N$ depends on the average ensemble weights (i.e., $\mathcal{V}_i$), since $L_\mathcal{D}(h_i)$ and $T$ are fixed in

collaboration. The bound in Proposition 1 suggests that different $\{\mathcal{V}_i\}_{i \in N}$ could lead to different $L_{\mathcal{D}}(h_N)$. Therefore, $\mathcal{V}_i$ indeed quantifies the influence of party $i$ to $L_{\mathcal{D}}(h_N)$ or its contribution to the ensemble in collaboration.

As parties would be primarily interested in receiving high-quality ensemble predictions, their contributions to an inferior ensemble are meaningless. For example, in a two parties' collaboration (e.g., $A$ and $B$) where $L_{\mathcal{D}}(h_A) = 0$ and $L_{\mathcal{D}}(h_B) = 1$, if $\phi_B = 1$ by a certain ensemble method, it does not imply that party $B$ possesses a better model. Additionally, party $B$ cannot benefit from the wrong ensemble predictions even if it receives all rewards. If a proper ensemble method (e.g., the optimal ensemble method in Sec. 4.1) is used, $\mathcal{V}_i$ should be decreasing with respect to $L_{\mathcal{D}}(h_i)$. It implies that a model with a large generalization error is assigned with a low average ensemble weight, such that the ensemble generalization error $L_{\mathcal{D}}(h_N)$ will be small and the ensemble predictions will be helpful in improving the performance. Thus, for a high-quality ensemble, a larger $\mathcal{V}_i$ implies a more valuable model with a smaller generalization error, which shows the usefulness of our valuation function $\mathcal{V}$. We have also empirically observed the validity of $\mathcal{V}$ in Sec. 6.

## 5 INCENTIVIZED BLACK-BOX MODEL SHARING

We identify that fairness and IR are the primary incentives for self-interested parties. In this section, we first introduce the incentive definitions and then show how our reward and payoff allocation scheme ensures fairness based on *contributions* and *payments*. Notably, our allocation scheme is also applicable to other valuation functions (e.g., $\mathcal{V}$ in Tay et al. (2022)) when the reward can be replicated (i.e., digital goods). Furthermore, we will theoretically examine how $\epsilon$-IR is satisfied.

To precisely describe the incentives, some notations are required: let $r_i \in \mathbb{R}_+$ denote the numerical *value* of party $i$'s reward, and let $p_i \in \mathbb{R}_+$ denote the value of the monetary payment that party $i$ makes to buy additional rewards. The payment $p_i$ is freely chosen by each party and it only flows to other parties (i.e., $N \setminus \{i\}$) as compensation. Let $r_i^+ \in \mathbb{R}_+$ denote the value of the additional rewards that party $i$ bought. Let $p_i^+ \in \mathbb{R}_+$ denote the value of monetary compensation that party $i$ receives from other parties' payments. Finally, let $L_{\mathcal{D}}(h_i')$ represent the generalization error of party $i$'s new model trained with additional ensemble predictions. Refer to Fig. 1 for a visual demonstration.

T1 **Shapley Fairness**. The rewards $\{r_i\}_{i \in N}$ and the monetary gain $\{r_i^+ - p_i + p_i^+\}_{i \in N}$ should be *Shapley Fair*: $\exists k_1, k_2 > 0, \text{s.t. } \forall i \in N, (r_i = k_1 \phi_i) \wedge (r_i^+ - p_i + p_i^+ = k_2 \phi_i)$.

T2 $\epsilon$-**Individual Rationality**. Each party receives a reward that at least improves its model performance and an additional reward that is at least as valuable as its payment: $\exists \epsilon > 0, \text{s.t. } \forall i \in N, (L_{\mathcal{D}}(h_i') - \epsilon \leq L_{\mathcal{D}}(h_i)) \wedge (r_i^+ \geq p_i)$.

T3 **Weak Efficiency**. At least one party in the collaboration should receive a reward that is as valuable as all ensemble predictions: $\exists i \in N, \text{s.t. } (r_i = \mathcal{V}_N) \wedge (p_i^* = 0)$, where $p_i^*$ denotes the maximal payment that party $i$ can make to exchange for the remaining rewards (i.e., $\mathcal{V}_N - r_i$).

Intuitively, T1 suggests that if a party $i$ has a larger contribution, both its reward $r_i$ and monetary payment gain $r_i^+ - p_i + p_i^+$ should be proportional to its Shapley value to ensure fairness. For instance, if two parties $i$ and $j$ make the same payment (i.e., $p_i = p_j$) and party $i$ has a larger contribution (i.e., $\phi_i > \phi_j$), it is only fair for party $i$ to receive more monetary gain (i.e., $r_i^+ - p_i + p_i^+ > r_j^+ - p_j + p_j^+$). ($r_i^+ \geq p_i$) means it only makes sense to pay if the additional reward is more valuable than the payment. Besides, a self-interested party desires the strict IR (i.e., $L_{\mathcal{D}}(h_i') \leq L_{\mathcal{D}}(h_i)$) before it joins the collaboration, which means the party always wants to be better off. However, it is generally challenging to analyze the generalization error (Jiang et al., 2020). Thus, we propose a relaxed version, namely $\epsilon$-IR, for theoretical analysis. We also notice the similar $\epsilon$-IR definitions from Roth & Shorrer (2017); Mounir et al. (2018). Additionally, for the *weak efficiency* concept, we adopt it from Sim et al. (2020), to ensure full utilization of the rewards and eliminate waste. From an implementation perspective, T3 enables a unique $k_1, k_2$ to satisfy both T1 and T3, discussed later.

### 5.1 FAIR REWARD AND PAYOFF ALLOCATION

A party $i$ can make payment $p_i$ to acquire additional reward $r_i^+$ to "top-up" its received reward $r_i$, as long as $r_i < \mathcal{V}_N$. However, to continue to ensure fairness it is necessary to determine a fair exchange

ratio between the payment $p_i$ and the additional rewards $r_i^+$. To this end, we first present a new game that quantifies the fair value of the monetary gain (i.e., $r_i^+ - p_i + p_i^+$):

**Definition 5.1** (Fair Replication Game). *The fair replication game, for a given Shapley vector $\{\phi_i\}_{i=1}^n$ and a payment vector $\{p_i\}_{i=1}^n$, is $G^p = (N, \mathcal{V}^p, \phi_i)$ and the characteristic function $\mathcal{V}^p$ is:*

$$\mathcal{V}^p(C) = \left(\sum_{i \in C} \phi_i\right) \times \gamma, \ \forall C \subseteq N \ where \ \gamma = \sum_{j \in N} p_j / (\mathcal{V}_N - \phi_j) \ .$$

FRG shows that the total monetary gain is $\mathcal{V}^p(N) = \mathcal{V}_N \times \gamma$, where the formulation $\left(\sum_{i \in C} \phi_i\right) \times \gamma$ of $\mathcal{V}^p$ suggests that each party shares a monetary gain of $\phi_i \times \gamma$ from $\mathcal{V}^p(N)$. Especially, $\gamma$ help establish a fair exchange ratio $\mathcal{V}_N / (\mathcal{V}_N - \phi_i)$ between $p_i$ and $r_i^+$, shown later in Theorem 1. This exchange ratio is also intuitive, as it implies that, for the same reward, a higher-contributing party will need less payment to acquire the reward, because the party has contributed more to the ensemble prediction (i.e., the source of the reward).

Intuitively, the overall gain of party $i$ in the collaboration is quantified by the sum of its reward $r_i$ and monetary gain $(r_i^+ - p_i + p_i^+)$. To analyze such value in the cooperative setting, we consider the combined game of both the weighted ensemble game $G$ and the fair replication game $G^p$. We denote the corresponding Shapley value $\phi_i(\mathcal{V} + \mathcal{V}^p)$ of party $i$ in the combined game as $u_i$, termed the *utility*. It represents the fair division of the overall gain from the combined game. We show in Theorem 1 that $u_i$ specifies both reward allocation and payoff flow, which exactly represents our reward and payoff allocation scheme.

**Theorem 1.** *Given $\mathcal{V}$ and $\mathcal{V}^p$, the utility $u_i := \phi_i(\mathcal{V} + \mathcal{V}^p)$ for each party $i \in N$ can be decomposed as $u_i = r_i + r_i^+ + p_i^+ - p_i$. Specifically,*

$$u_i = r_i + \underbrace{\frac{\mathcal{V}_N \times p_i}{\mathcal{V}_N - \phi_i}}_{r_i^+} + \underbrace{\sum_{j \in N \setminus \{i\}} \frac{\phi_i \times p_j}{\mathcal{V}_N - \phi_j}}_{p_i^+} - p_i$$

*where $r_i = \phi_i$ is the Shapley value from the game $G$. Also, it satisfies*
*(a) payoff balance: $\sum_{i \in N}(p_i^+ - p_i) = 0$, (b) dummy payment: $\forall C \subseteq N \setminus \{i\}$, $\mathcal{V}(C \cup i) = \mathcal{V}(C) \Rightarrow u_i = 0$, $r_i^+ = p_i$, (c) semi-symmetry: $\forall C \subseteq N \setminus \{i, j\}$, $\mathcal{V}(C \cup i) = \mathcal{V}(C \cup j) \Rightarrow u_i = u_j$, and (d) strict monotonicity: $(\exists j \in N \ p_j' > p_j) \wedge (\forall k \in N \ p_k' \geq p_k) \Rightarrow \forall i \in N \ u_i' > u_i$.*

**Remark 1.** *The four properties indicate: (a) The host does not collect any payment; instead, payments made by any party only serve as compensations to other parties; (b) A dummy party (i.e., where $\phi_i = 0$) receives a utility of zero; (c) When two parties contribute equally, the payments they make do not influence their relative utility values, which shows a fair trade-off between payoffs and rewards; (d) Should a single party increase its payment, it results in a rise in utility for all parties.*

The proof is given in App. C.2. Allocating rewards and payoffs based on *utility* directly satisfies the fairness incentive T1, where party $i$ receives a reward of value $r_i + r_i^+ = r_i + \mathcal{V}_N \times p_i / (\mathcal{V}_N - \phi_i)$, and a payoff of value $p_i^+ - p_i = \sum_{j \in N \setminus i} \frac{\phi_i \times p_j}{\mathcal{V}_N - \phi_j} - p_i$. Especially, there are infinitely many $k_1$ such that $r_i = k_1 \phi_i$ to satisfy T1. Enforcing T3, by using the scaled reward $r_i = (\phi_i / \phi^*) \times \mathcal{V}_N$ where $\phi^* = \max_{i \in N} \phi_i$, leads to a unique $k_1 = \mathcal{V}_N / \phi^*$ that jointly satisfies T1 and T3. It allocates the maximal reward $\mathcal{V}_N$ to the most-contributing party. This scaled reward is the result of $u_i := \phi_i(k_1 \mathcal{V} + \mathcal{V}^p)$ in Theorem 1. As the maximum value of reward cannot exceed $\mathcal{V}_N$, the maximal payment that each party can make to increase its reward is $p_i^* = (\phi^* - \phi_i)(\mathcal{V}_N - \phi_i)/\phi^*$. When all parties make their maximal payments, each party $i \in N$ receives all ensemble predictions of value $\mathcal{V}_N$ and a payoff of $n\phi_i - \mathcal{V}_N$. To elaborate on T2, as the exchange ratio $\mathcal{V}_N / (\mathcal{V}_N - \phi_i)$ between $p_i$ and $r_i^+$ is greater than 1, $r_i^+ \geq p_i$ is always satisfied. The proofs of the above incentive guarantee are given in App. C.6. Notably, our allocation scheme does not depend on any particular ensemble method. Given any contribution measure (i.e., $\phi_i$), our allocation scheme satisfies T1&T3, as it essentially establishes a fair trade-off between rewards and payoffs.

**Reward Realization** After determining the rewards $\{r_i\}_{i=1}^n$ and collecting the payments $\{p_i\}_{i=1}^n$, the host can realize the reward and payoff allocation. The final payoff for party $i$ is its compensation offsetting its payment: $p_i^+ - p_i$. When $p_i^+ - p_i > 0$, party $i$ receives monetary profit. The reward $r_i + r_i^+$ is realized as a set of data of size $T_i$ uniformly randomly sampled from the ensemble predictions $\{x_t, h_N(x_t)\}_{t=1}^T$. We then have $T_i := (r_i + r_i^+) \times T = (\phi_i / \phi^* + p_i / (\mathcal{V}_N - \phi_i)) \times \mathcal{V}_N \times T$.

Note that $\phi_i = \sum_{x \in U} \beta_{i,x}/T$ shown in Sec. 4.2. This shows a party with a larger average ensemble weight and payment will be rewarded with more ensemble predictions (i.e., a larger $T_i$).

## 5.2 GUARANTEEING $\epsilon$-INDIVIDUAL RATIONALITY

As $r_i^+ \geq p_i$ is always satisfied by our allocation scheme above, we will only focus on how $L_{\mathcal{D}}(h_i') - \epsilon \leq L_{\mathcal{D}}(h_i)$ is satisfied to guarantee $\epsilon$-IR. The $\epsilon$-IR incentive ensures that parties participating in collaboration will not experience more than $\epsilon$ "regret" of generalization error. Therefore, the regret $\epsilon$ has to be minimized to incentivize participation. We next give our exact expression of $\epsilon$. Denote $h_i' \in \mathcal{H}$ as the empirical minimizer of the risk $\hat{L}_{\mathcal{D}_i'}(h) := (1 - \alpha_i)\hat{L}_{\mathcal{D}_i}(h) + \alpha_i \hat{L}_{\mathcal{D}}(h, h_N)$ for each party $i \in N$ and $\alpha_i \in [0,1]$ the mixing value for balancing training data. Therefore, $L_{\mathcal{D}}(h_i')$ is the generalization error of model $h_i'$ trained on its source data $\mathcal{D}_i$ and ensemble predictions of size $T_i$. By leveraging technical results from domain learning theory (Ben-David et al., 2010) and utilizing the *ensemble domain* $\langle \mathcal{D}, h_N \rangle$, we provide a specific result of $\epsilon$-IR in Proposition 2.

**Proposition 2.** *Let $\mathcal{H}$ be a hypothesis space of VC dimension $d$. Given $m_i$ the data size of $\mathcal{D}_i$, and a distribution divergence measure $d_{\mathcal{H}\Delta\mathcal{H}}(\mathcal{D}_i, \mathcal{D}) := 2\sup_{h,h' \in \mathcal{H}} |L_{\mathcal{D}_i}(h, h') - L_{\mathcal{D}}(h, h')|$, with probability at least $1 - n\delta$, $\epsilon$-IR is satisfied such that $\forall i \in N$ $L_{\mathcal{D}}(h_i') - \epsilon \leq L_{\mathcal{D}}(h_i)$ with $\epsilon = \max_{i \in N} \epsilon_i$ and*

$$\epsilon_i = 4\sqrt{2d\log(2(T_i + m_i + 1)) + 2\log(\frac{8}{\delta})}\sqrt{\frac{\alpha_i^2}{T_i} + \frac{(1 - \alpha_i)^2}{m_i}}$$
$$+ (1 - \alpha_i)d_{\mathcal{H}\Delta\mathcal{H}}(\mathcal{D}_i, \mathcal{D}) + 2\alpha_i L_{\mathcal{D}}(h_N).$$

The proof is given in App. C.3. As $\epsilon = \max_{i \in N} \epsilon_i$, to minimize $\epsilon$ it is equal to minimize $\epsilon_i$. Thus, a smaller $\epsilon_i$ for all $i \in N$ indicates a stronger $\epsilon$-IR guarantee. Proposition 2 shows that $\epsilon_i$ mainly depends on the ensemble error $L_{\mathcal{D}}(h_N)$, the mixing value $\alpha_i$, and its reward size $T_i$, as $d_{\mathcal{H}\Delta\mathcal{H}}(\mathcal{D}_i, \mathcal{D})$ is fixed. For a given collaboration, $L_{\mathcal{D}}(h_N)$ (due to ensemble method) and $T_i$ (due to fixed $\phi_i$ and $p_i$) are also fixed. We can write $\epsilon_i(\alpha_i)$ as a function of $\alpha_i$. When $\alpha_i = 1$, it implies $h_i'$ is only trained on the ensemble predictions, and $\epsilon_i(1)$ mainly depends on $L_{\mathcal{D}}(h_N)$ the error of ensemble predictions; when $\alpha_i = 0$, $h_i'$ is the same as $h_i$, and $\epsilon_i(0)$ mainly depends on $d_{\mathcal{H}\Delta\mathcal{H}}(\mathcal{D}_i, \mathcal{D})$ the distribution divergence. As $\epsilon_i(\alpha_i)$ varies with different $\alpha_i$, we will next analyze the minimum of $\epsilon_i(\alpha_i)$. To ease the notation, we omit the subscript $i$ and rewrite $\epsilon_i(\alpha_i)$ as:

$$\epsilon_i(\alpha) = B\sqrt{\alpha^2/T + (1 - \alpha)^2/m} + \alpha A + d_{\mathcal{H}\Delta\mathcal{H}}(\mathcal{D}_i, \mathcal{D}) \tag{1}$$

where $A = -d_{\mathcal{H}\Delta\mathcal{H}}(\mathcal{D}_i, \mathcal{D}) + 2L_{\mathcal{D}}(h_N)$, and $B = 4\sqrt{2d\log(2(T + m + 1)) + 2\log(\frac{8}{\delta})}$. Let $C$ denote the ratio $A^2/B^2$. We can find the optimal value $\alpha^*$ that minimizes $\epsilon_i(\alpha)$:

$$\alpha^* = \begin{cases} 1 & T > m(Cm - 1)^{-1} \\ \min\{1, \xi\} & T \leq m(Cm - 1)^{-1} \end{cases} \quad \text{where } \xi = \frac{T}{m + T}\left(1 + \frac{m\sqrt{C}}{\sqrt{m + T - CTm}}\right). \tag{2}$$

The proof is shown in App. C.4. First, $\alpha_i^*$ provides insights regarding the parameter choice in balancing the training data, which would serve practitioners well. If $m_i = 0$ or $T_i = 0$, then $\alpha_i^* = 1$ or 0 accordingly, which intuitively suggests that party $i$ should train its model with whatever data it has. If $T_i > m_i(C_i m_i - 1)^{-1}$, it implies that there are enough rewards (i.e., the ensemble predictions) that party $i$ gets from the collaboration, its source data should be ignored in training the model $h_i'$.

Besides, if $\alpha_i^*$ is used to minimizes $\epsilon_i$ for all $i$ in $N$, we then have the strongest $\epsilon$-IR guarantee where $\epsilon = \max_{i \in N} \epsilon_i$ is minimized. In the ideal case if $\epsilon_i = 0 \ \forall i \in N$, the strict IR (i.e., $L_{\mathcal{D}}(h_i') \leq L_{\mathcal{D}}(h_i)$) is satisfied; however, the minimum of $\epsilon_i$ is generally positive, and the strict IR is hard to achieve theoretically. We will later empirically show that the virtual regret $\epsilon$ is not needed and the strict IR is satisfied in Figs. 2, 7 and 10, which suggests parties will never be worse off from collaboration. We will also empirically show in Fig. 3 that $\alpha_i^*$ not only results in the strict IR, but also could bring the largest performance improvement (i.e., the strongest strict IR).

## 6 EXPERIMENTS AND DISCUSSION

**Valuation** To justify our valuation method, we examine the correlation between the average ensemble weight $\mathcal{V}_i$ ($\phi_i = \mathcal{V}_i$) and the generalization error $L_{\mathcal{D}}(h_i)$ of each party $i$. We perform

experiments on different datasets including MNIST (LeCun et al., 1998), CIFAR-10 (Krizhevsky et al., 2009), and SVHN (Netzer et al., 2011). The training dataset is divided into 5 random i.i.d. subsets, representing a hypothetical collaboration among $n = 5$ parties, with each party having access to a distinct (i.e., non-overlapping) subset of the data. Each party adopts a neural network with two fully connected layers to fit their respective data. $\mathcal{V}_i$ is calculated on a validation dataset of size 5000, and $L_{\mathcal{D}}(h_i)$ is estimated over the test dataset. When different ensemble methods (refer to App. B.2 for more details) are used in Table 1, the AVG method reports near zero correlation, as it is ineffective in identifying models with lower $L_{\mathcal{D}}(h_i)$. When the optimal ensemble method is used, Table 2 shows $\mathcal{V}_i$ and $L_{\mathcal{D}}(h_i)$ is highly correlated. This result helps validate Proposition 1 and helps justify our valuation method. We also examine our valuation function under the model/data heterogeneous settings and study how different ensemble methods affect both average ensemble weight $\mathcal{V}_i$ and the accuracy of ensemble predictions in App. D.1.

Table 1: Pearson correlation between $\mathcal{V}_i$ and $L_{\mathcal{D}}(h_i)$ on MNIST with different ensemble methods. The value is reported over 100 independent evaluations.

Table 2: Pearson correlation between $\mathcal{V}_i$ and $L_{\mathcal{D}}(h_i)$ on different datasets. The value is reported with the mean and standard error over 100 independent evaluations.

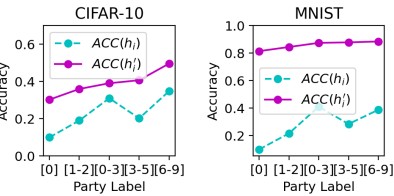

| Ensemble | Correlation |
|----------|-------------|
| AVG | -0.02±0.54 |
| MV | -0.24±0.52 |
| MWU | -0.24±0.56 |

| Dataset | Correlation |
|---------|-------------|
| MNIST | -0.72±0.35 |
| CIFAR-10 | -0.90±0.06 |
| SVHN | -0.83±0.11 |

Figure 2: Test accuracy (ACC) of five parties' models before and after incorporating fair ensemble prediction rewards.

$\epsilon$-**IR** To illustrate how rewards are fairly distributed to improve model performance, we experiment on MNIST and CIFAR-10, and follow the same experimental setting as (Nguyen et al., 2022) to partition the training dataset based on the class labels. A party, denoted as $[s-e]$ $(s \leq e)$, owns a subset of the training dataset that is labeled with classes $s, s+1, \ldots, e$. We consider 5 parties in $N$: [0], [1-2], [0-3], [3-5], and [6-9]. Each party uses a neural network with a single hidden layer to fit its training data and only shares the predictions for an unlabeled dataset of size $T = 5000$ sampled from the training data. With the optimal ensemble method, we obtain the Shapley value vectors $[0.087, 0.112, 0.200, 0.224, 0.377]$ on MNIST, and similarly $[0.078, 0.136, 0.195, 0.208, 0.383]$ on CIFAR-10, which are intuitively fair as the unique data is more valuable. This again *justifies* our valuation method. The ensemble predictions are then distributed according to our allocation scheme in Sec. 5.1 where we ignore the payment first. Fig. 2 shows that the test accuracy of all parties is improved after incorporating the ensemble predictions, i.e., $\forall i \in N, \text{ACC}(h_i') \geq \text{ACC}(h_i)$. In other words, we empirically achieve the strict IR (i.e., $L_{\mathcal{D}}(h_i') \leq L_{\mathcal{D}}(h_i)$). Although party [3-5] (i.e., $i = 4$) cannot achieve higher accuracy by itself due to fewer data than party [0-3], party [3-5]'s data uniqueness helps it have larger marginal contributions and achieve higher accuracy after receiving more rewards. This encourages collaboration among different organizations, particularly those with diverse and unique data. The optimal $\alpha^*$ is used in Fig. 2. Refer to App. D.2 for more results under different $T$. We will next show how $\alpha^*$ is closely related to $\epsilon$-IR. With the optimal ensemble method, $L_{\mathcal{D}}(h_N) = 0$. Through estimating $d_{\mathcal{H}\triangle\mathcal{H}}(\mathcal{D}_i, \mathcal{D})$ and the VC dimension $d$, we could find the optimal value $\alpha^*$ shown in Eqn. 2. We continue with the above experimental setting of MNIST, and use party [0-3] with a median Shapley value as an example. Fig. 3a shows $\epsilon_i(\alpha_i)$ changes with $\alpha_i$ and our calculated $\alpha_i^*$ is indeed at its minimum. We can observe from Fig. 3b that $\alpha_i$ affects the generalization error drop $L_{\mathcal{D}}(h_i') - L_{\mathcal{D}}(h_i)$, and our optimal $\alpha_i^*$ results in the largest drop, which implies the greatest model improvement. The best model performance should intuitively indicate the strongest $\epsilon$-IR guarantee, which is just reflected by Fig. 3c. As we should have $L_{\mathcal{D}}(h_i') - L_{\mathcal{D}}(h_i) - \epsilon_i \leq 0$ from $\epsilon$-IR, the least value of $(L_{\mathcal{D}}(h_i') - L_{\mathcal{D}}(h_i) - \epsilon_i)$ represents the strongest $\epsilon$-IR guarantee, and our $\alpha_i^*$ also captures this. Refer to App. D.2 for more results on CIFAR-10.

**Fair Allocation** We again follow the above experimental setting of MNIST and demonstrate the efficacy of our allocation scheme. From Fig. 4, we observe the strict monotonicity property defined in Theorem 1: when only one party makes the payment (e.g., $p_1$), the utility of each party linearly increases with $p_1$ and is proportional to contributions, which ensures fairness when other parties acquire more rewards with payments. We next compare our allocation scheme against Nguyen et al.

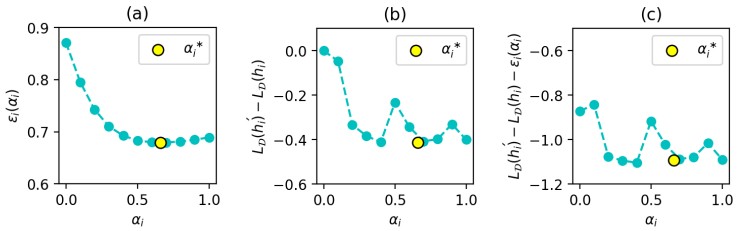 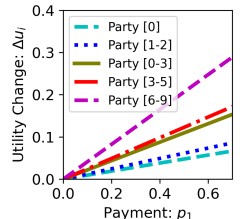

Figure 3: (a) The function value of $\epsilon_i(\alpha_i)$ and its minimum $\epsilon_i(\alpha_i^*)$, (b) the generalization error drop $L_{\mathcal{D}}(h_i') - L_{\mathcal{D}}(h_i)$ with different $\alpha_i$, and (c) the $\epsilon$-IR guarantee quantified by $L_{\mathcal{D}}(h_i') - L_{\mathcal{D}}(h_i) - \epsilon_i(\alpha_i)$ with different $\alpha_i$.

Figure 4: Utility gains of five parties when only party [0] makes payment.

(2022), and to do so we assume every party wants to maximize its reward, in order to specifically compare with the baseline. Note that this assumption is only to enable the comparison, as our allocation allows parties to freely choose whether to accept $p_i^+$ or use $p_i^+$ to further increase its reward (i.e., our allocation does *not* require this assumption). We first consider the example where only party [0] makes a payment of value 0.5 (i.e., $p_1 = 0.5$). Fig. 5a shows our allocation scheme satisfies Shapley fairness while the baseline does not. As the corresponding payoff (i.e., $p_i^+ - p_i$) vectors for both methods are the same $[-0.5, 0, 0, 0, 0.5]$, we observe from Fig. 5b that the baseline method violates fairness because of over distributing rewards. We then consider the example where only party [0] and [0-3] make payments where $p_1 = 0.5$ and $p_3 = 0.1$. The payoff flows for both methods are shown Fig. 5c (baseline) and Fig. 5d (ours). As party [6-9] is the most contributing party and T3 ensures it receives the maximum reward (i.e., $r_5 = \mathcal{V}_N$), it intuitively has no need to make any payments, which is justified by our payoff flow in the last row of Fig. 5d. However, the payoff flow of the baseline method in Fig. 5c requires party [6-9] to unnecessarily have an outgoing payment flow. When parties have no budget constraint, the utility in Nguyen et al. (2022) could satisfy Shapley fairness (i.e., $\exists k > 0$, s.t. $\forall i \in N, u_i = k\phi_i$), and its payoff flow is the same as ours. However, its utility cannot satisfy Shapley fairness when budget constraints are applied. In contrast, our allocation scheme ensures fair utility under any budget constraint. Refer to App. D.3 for more examples demonstrating our fair utility.

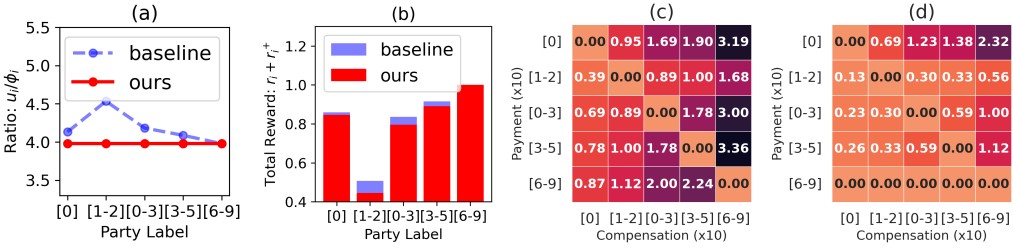

Figure 5: (a) The ratio of the utility over the Shapley value of each party when only party [0] makes a payment, (b) the total received reward of each party when only party [0] makes a payment, (c) the payoff flow of baseline method when party [0] and [0-3] make payments, and (d) the payoff flow of our method when party [0] and [0-3] make payments.

## 7 CONCLUSION

In this paper, we have introduced the novel collaborative framework of incentivized black-box model sharing as an appealing choice for collaborative learning. We perform a fair contribution evaluation of black-box models using the average ensemble weight and demonstrate its validity both theoretically and empirically. We also derived a concise and closed-form solution to distribute rewards and payoffs while guaranteeing fairness. Finally, we show $\epsilon$-IR is satisfied theoretically and strict IR can be achieved empirically. For future work, we will consider the collaboration setting where different parties have different target domains and parties may not always respond to the host's queries.

## REPRODUCIBILITY STATEMENT

To ensure the transparency and reproducibility of our work, we presented a detailed explanation of our valuation method in Sec. 4, the fair allocation scheme in Sec. 5.1, and the $\epsilon$-IR in Sec. 5.2. All theoretical results were detailed with clarity in our main paper, including assumptions made for each algorithm. Complete proofs of all our claims are also available in App. C. For the empirical results of this study, we have included specific experimental settings in the main paper and App. D, and have also made the source code of our model openly accessible in the supplementary material (i.e., the zip file).

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

## A    FURTHER CLARIFICATIONS

(This subsection of summary is added during the rebuttal)

**Summary**    This paper proposes a theoretical framework for incentivized black-box model sharing with a 2-stage mechanism as illustrated in Fig. 1. Below we describe clearly the steps of the proposed mechanism.

In **stage 1** (in Sec. 4),

1. Each party $i \in N$ has an already trained multi-class classifier $h_i$ on its own source distribution $\mathcal{D}_i$, and is interested in maximizing the performance on the same target distribution $\mathcal{D}$.

2. Each party $i$ sends its prediction $h_i(x)$ to a trusted host for each $x \in U$ where $U \sim \mathcal{D}$ is the distillation dataset whose size $|U| = T$.

3. The host produces the ensemble prediction $h_N(x) = \sum_{i=1}^{n} \beta_{i,x} h_i(x)$ for each $x \in U$ where $\beta_{i,x}$ is the ensemble weight of $h_i(x)$ determined by a *given* choice of the ensemble such as AVG, MV, MWU or MV (see how to determine $\beta_{i,x}$ in App. B.2).

4. The fair contribution of each party $i$ is defined as the Shapley value $\phi_i$, using the valuation function $\mathcal{V}$ in Def. 4.1.

In **stage 2** (in Sec. 5),

1. Each party $i$ can make an additional monetary payment $p_i \in \mathbb{R}_+$ through the trusted host to other parties to obtain additional rewards (of value) $r_i^+ \in \mathbb{R}_+$. Each party $i$ decides on its own the exact amount $p_i$.

2. Once all parties have finalized their payments $\{p_i\}_{i=1}^{n}$, each party $i$ will receive a total reward of value $(r_i + r_i^+)$, which is realized as an i.i.d. subsampled set $U_i$ of the *ensemble predictions* $\{x_t, h_N(x_t)\}_{t=1}^{T}$.

3. The payments made by other parties (i.e., $N \setminus \{i\}$) are allocated by the host to party $i$ as its monetary compensation $p_i^+$. The exact allocation, namely the value of $(r_i + r_i^+)$ and $p_i^+$ are fully specified by Theorem 1.

### A.1    LIMITATIONS

(App. A.1 is added during the rebuttal)

One limitation of this work is that the model valuation method only applies to the weighted sum ensemble, which may limit its potential application to other black-box model sharing schemes where different ensemble methods are used. Besides, our valuation method relies on the ensemble method used in the collaboration to determine the ensemble weight $\beta_{i,x}$. If the ensemble method is defective, $\beta_{i,x}$ may not reflect the true quality of the black-box model, resulting in inaccurate ensemble predictions. Future works may consider improving the ensemble method by utilizing our optimal ensemble to learn model surrogates with a small set of labeled data. Those model surrogates may serve as estimators to obtain accurate $\beta_{i,x}$.

### A.2    Q&A

**Q1**: What is the access to the host and each party?
**A1**: In the black-box model sharing, each party does not know any information about the others, and the host only has access to the distillation dataset $U$ and the predictions (i.e., $h_i(x)$) from all parties. Our theoretical results do not have assumptions on the difference across the source distribution $\mathcal{D}_i$. We assume that each party wants to perform well on the same test dataset (i.e., the target distribution $\mathcal{D}$), but the ground truth (i.e., $f(x) \ \forall x \sim \mathcal{D}$) of the test dataset is unknown. Please also refer to Fig. 1 for the overview of the framework of incentivized black-box model sharing.

**Q2**: What would be a completely described motivating example of incentivized black-box model sharing?
**A2**: For instance, hospitals with varying domain expertise and locations may aim to optimize patient

treatment collectively while maintaining data and model privacy. This can be achieved through black-box model sharing. A trusted host, like a government agency, can use a jointly accepted dataset (i.e., $U$) to distill individual model knowledge (i.e., acquire prediction $h_i(x)$). Using an appropriate ensemble method, the host can determine each hospital's contribution (i.e., $\phi_i$). Lesser contributing hospitals can further compensate others through payments (i.e., $p_i$) for more rewards (i.e., $r_i^+$). The host will distribute the ensemble predictions as rewards based on contributions and payments.

**Q3**: How should we choose the ensemble weight $\beta_{i,x}$ to satisfy both fairness and $\epsilon$-IR?
**A3**: As stated in Sec. 4, the ensemble weight $\beta_{i,x}$ is specified by the ensemble method which is used by the host to aggregate predictions. It only quantifies the contribution of any party to the ensemble predictions (i.e., the contribution to collaboration). However, the contribution to a bad ensemble might be meaningless, as the bad-quality ensemble predictions will not help improve the model's performance. In general, to have a good-quality ensemble prediction, the average ensemble weight $\sum_{x \in U} \beta_{i,x}/T$ should be small when its generalization error $L_{\mathcal{D}}(h_i)$ is large, as suggested in Proposition. 1. As the ensemble weight $\beta_{i,x}$ is only a measure of contribution, it does not affect the fairness. In fact, fairness is always satisfied by our allocation scheme. For $\epsilon$-IR, the ensemble weight of party $i$ will affect its $T_i$, and further influence $\epsilon$. Thus, the larger the $\sum_{x \in U} \beta_{i,x}/T$, the smaller the $\epsilon$.

**Q4**: Why can we value the payment and the reward together?
**A4**: Consider an example of collaboration where there are 3 parties with the monetary budget of $100, $500, and $1000 respectively. Inspired by Han et al. (2023) which values the rewards using a monetary value, we denote any monetary currency as $s$ (e.g., $200), and further denote $\nu$ an exchange ratio such that it projects the monetary currency $s$ into the positive real domain $\mathbb{R}_+$ to align the reward domain and the monetary domain. To normalize the scale, we need to have $s/S = p_i/\mathcal{V}_N$, where $S$ is the normalization of the monetary domain, and $\nu = \mathcal{V}_N/S$. For example, let $S$ be the total budget of all parties (i.e., $1600). If $s =$200, $p_i = s\nu = 0.125$. Note that we have $\mathcal{V}_N = 1$ in our setting. As $r_i \in \mathbb{R}_+$ and $0 \leq r_i \leq \mathcal{V}_N$, we can value the monetary payment and the reward together.

**Q5**: How is the reward related to the ensemble predictions?
**A5**: As we have defined $T_i := (r_i + r_i^+) \times T = (\phi_i/\phi^* + p_i/(\mathcal{V}_N - \phi_i)) \times \mathcal{V}_N \times T$ in Sec. 5.1, the reward value $(r_i + r_i^+) \in \mathbb{R}_+$ is linearly mapped to the number of ensemble predictions. While one might use other function $g$ to map $(r_i + r_i^+)$ to $T_i$, as long as $\mathcal{V}_N$ is mapped to $T$, our linear mapping is more intuitive since fairness itself suggests the proportionality. Denote $U_i$ the set of ensemble predictions that party $i$ receives. In our experiments, we also impose that $U_i \subseteq U_j \ \forall i, j \in N \ s.t. \ \phi_i \leq \phi_j$, which leads to strictly the same rewards for equally contributing parties and strictly higher rewards for parties with larger contributions.

**Q6**: How meaningful is the additive structure of $\mathcal{V}$ and $\mathcal{V}^p$?
**A6**: We identify the additive structure in existing ensemble methods (Feng et al., 2021; Chang et al., 2021; Lin et al., 2020), where the additivity is naturally observed as the ensemble prediction is a weighted sum of individual predictions. We further exploit the additivity to define the $\mathcal{V}$, such that the Shapley value can be efficiently obtained. Instead of directly measuring a model quality, our $\mathcal{V}$ quantifies the importance/weights of predictions from coalition $C$ in producing ensemble predictions. We notice a recent work Drungilas et al. (2023) proposed to determine the ensemble weight $\beta_{i,x}$ by measuring the contribution of a learner to an ensemble using Shapley value. Conversely, our valuation method directly uses the ensemble weight as the contribution, where the ensemble weight is the Shapley value in our formulation. Besides, our $\mathcal{V}$ is independent of the ensemble method used by the host. If a proper ensemble method is used, $\mathcal{V}_i$ then implicitly reflects the quality of a model. We have validated it with theoretical analysis in Proposition. 1 and empirical results in Table. 1&2. Besides, our formulation of $\mathcal{V}^p$ is also meaningful, as its Shapley solution characterizes the fair trade-off between reward and payoff, and also specifies the payoff flow as shown in Theorem. 1. It also has nice properties (i.e., (a-d) in Theorem. 1) that are meaningful in promoting collaborations.

# B  ADDITIONAL DESCRIPTION

## B.1  SHAPLEY VALUE

In the realm of cooperative game theory, the Shapley value has emerged as a primary solution concept to ensure fairness when allocating resources among a group of individuals. With its interpretation

as an expected marginal contribution of each player, the Shapley value is the unique solution that satisfies four significant properties as elaborated below.

**Efficiency**. All of the collaborative gain $\mathcal{V}_N$ is distributed to the parties: $\sum_{i \in N} \phi_i(\mathcal{V}) = \mathcal{V}_N$.

**Symmetry**. Parties with equal marginal contributions to any coalition in the collaboration receive the same gain (i.e., reward/payoff): $\forall C \subseteq N \setminus \{i,j\}$ $\mathcal{V}(C \cup i) = \mathcal{V}(C \cup j) \Rightarrow \phi_i(\mathcal{V}) = \phi_j(\mathcal{V})$.

**Dummy Party**. The dummy parties receives a gain of zero: $\forall C \subseteq N \setminus \{i\}$ $\mathcal{V}(C \cup i) = \mathcal{V}(C) \Rightarrow \phi_i(\mathcal{V}) = 0$.

**Linearity**.If two coalition games described by valuation functions $\mathcal{V}$ and $\mathcal{V}^p$ are combined, then the distributed gains should correspond to the gains derived from $\mathcal{V}$ and the gains derived from $\mathcal{V}^p$: $\phi_i(\mathcal{V} + \mathcal{V}^p) = \phi_i(\mathcal{V}) + \phi_i(\mathcal{V}^p)$, $\phi_i(\beta\mathcal{V}) = \beta\phi_i(\mathcal{V})$, $\forall i \in N$, $\forall \beta \in \mathbb{R}$.

==(The two more properties of Shapley value are newly added during the rebuttal.)==

It is also shown in Sim et al. (2020) that Shapley value also satisfies two more properties:

**Strict Desirability**. If the marginal contribution of party $i$ to at least a coalition is higher than that of party $j$, but the reverse is not true, then party $i$ should receive a higher gain than party $j$: For all $i, j \in N$ s.t. $i \neq j$,

$$(\exists B \subseteq N \setminus \{i,j\}\ \mathcal{V}_{B \cup \{i\}} > \mathcal{V}_{B \cup \{j\}}) \wedge (\exists C \subseteq N \setminus \{i,j\}\ \mathcal{V}_{C \cup \{i\}} \geq \mathcal{V}_{C \cup \{j\}}) \Rightarrow \phi_i > \phi_j.$$

**Strict Monotonicity.**. If the marginal contribution of party $i$ to at least a coalition improves (e.g., by sharing more accurate predictions), *ceteris paribus*, then party $i$ should receive a higher gain than before: Let $\{\mathcal{V}_C\}_{C \in 2^N}$ and $\{\mathcal{V}'_C\}_{C \in 2^N}$ denote two sets of values over all coalitions $C \subseteq N$, and $\phi_i$ and $\phi'_i$ be the corresponding values of collaborative gain received by party $i$. For all $i \in N$,

$$(\exists B \subseteq N \setminus \{i\}\ \mathcal{V}'_{B \cup \{i\}} > \mathcal{V}_{B \cup \{i\}}) \wedge$$
$$(\forall C \subseteq N \setminus \{i\}\ \mathcal{V}'_{C \cup \{i\}} \geq \mathcal{V}_{C \cup \{i\}}) \wedge$$
$$(\forall A \subseteq N \setminus \{i\}\ \mathcal{V}'_A \geq \mathcal{V}_A) \wedge (\mathcal{V}'_N \geq \phi_i) \Rightarrow \phi'_i > \phi_i.$$

## B.2 ENSEMBLE METHOD

We introduce several ensemble methods that we consider in our experiments for prediction aggregation. Generally, an ensemble method does not require knowing the precision of individual models on any data. The host that facilitates the ensemble distillation only knows the prediction $h_i(x)$ of each agent given a data $x$. A special case is made for the optimal ensemble method, where the labels of $U$ are assumed to be known by the host.

==(The optimization objective of optimal ensemble is updated during the rebuttal, where we include a regularization term.)==

**Optimal Ensemble**. We introduce an ideal ensemble method to avoid the choice of different ensemble methods in our main text. When the ground truth of $U$ is known by the host, the optimal ensemble weight $\{\beta^*_{i,x}\}_{i=1}^n$ for $x \in U$ is given by solving the following linear optimization problem:

$$\underset{\beta^*_{i,x} \in [0,1], \forall i \in N.}{\text{minimize}} \quad |f(x) - \sum_{i=1}^n \beta^*_{i,x} h_i(x)| + \lambda \sum_{\substack{i,i' \in N \\ i \neq i'}} (|\beta_{i,x} - \beta_{i',x}|)^{(2 - \|h_i(x) - h_{i'}(x)\|_1)}$$

$$\text{subject to} \quad \sum_{i=1}^n \beta^*_{i,x} = 1$$

where $\lambda$ is a parameter that controls the importance of the regularization term, and the hyperparameter 2 in $(2 - \|h_i(x) - h_{i'}(x)\|_1)$ is because the L-1 norm of the difference is upper bounded by 2. With the regularization, we can ensure that the optimal weight $\beta^*_{i,x}$ for different parties are close when their prediction $h_i(x)$ are similar.

**Average (AVG)**. It is the simplest ensemble method where the predictions from individual models are averaged as $h_N(x) = n^{-1} \sum_{i=1}^n h_i(x)$ where each party has $\beta_{i,x} = 1/n$.

**Majority Vote (MV)**. The class or label with the greatest number of votes is chosen as the output of MV, representing the consensus decision of the ensemble. Assume there are $c$ models that vote for

the consensus prediction, and denote such parties as coalition $C$. Then, we have $\beta_{i,x} = 1/c \; \forall i \in C$ and $\beta_{i,x} = 0 \; \forall i \in N \setminus C$.

**Knowledge Vote (KV)**. The method KV was proposed (Feng et al., 2021) to assign low ensemble weights to irrelevant and malicious models. The idea is that if a certain consensus prediction is approved by more parties with high confidence (e.g., $> 0.9$), then it will be more likely to be the true label. Assuming there are $c$ models that vote for the consensus prediction, and denote such parties as coalition $C$. Then, we have $\beta_{i,x} = 1/c \; \forall i \in C$ and $\beta_{i,x} = 0 \; \forall i \in N \setminus C$.

**Multiplicative Weight Update (MWU)**. We adopt MWU (Blum et al., 2021) to adaptively adjust the ensemble weight for each prediction, where the weight update and aggregate computation are given as:

$$\beta_{i,x}^{t+1} = -\log\left(\frac{\|h_N^t(x) - h_i(x)\|_2}{\sum_{i=1}^n \|h_N^t(x) - h_i(x)\|_2}\right)$$

$$h_N^{t+1}(x) = \sum_{i=1}^n \beta_{i,x}^{t+1} h_i(x).$$

Our study shows how different ensemble methods affect both the average ensemble weight $\mathcal{V}_i$ and the accuracy of ensemble predictions in App. D.1.

# C  PROOFS

## C.1  PROOF OF PROPOSITION 1

For our analysis, we assume the loss function defined as $\ell(h(x), f(x)) = |h(x) - f(x)|$, which is convex. Note that $\sum_{i=1}^n \beta_{i,x} = 1$ and $\beta_{i,x} \geq 0$. We have:

$$
\begin{aligned}
L_{\mathcal{D}}(h_N) &= \mathbb{E}_{x \sim \mathcal{D}}\left[\left|\sum_{i=1}^n \beta_{i,x} h_i(x) - f(x)\right|\right] \\
&= \mathbb{E}_{x \sim \mathcal{D}}\left[\sum_{i=1}^n |\beta_{i,x}(h_i(x) - f(x))|\right] \qquad (h_i(x) = 1) \\
&= \sum_{i=1}^n \mathbb{E}_{x \sim \mathcal{D}}[\beta_{i,x}|h_i(x) - f(x)|] \\
&= \sum_{i=1}^n \left\{\mathbb{E}_{x \sim \mathcal{D}}[\beta_{i,x}] L_{\mathcal{D}}(h_i) + \mathbb{E}_{x \sim \mathcal{D}}[(\beta_{i,x} - \mathbb{E}_{\mathcal{D}}[\beta_{i,x}])(|h_i(x) - f(x)| - L_{\mathcal{D}}(h_i))]\right\} \\
&= \sum_{i=1}^n \mathbb{E}_{x \sim \mathcal{D}}[\beta_{i,x}] L_{\mathcal{D}}(h_i) + \sum_{i=1}^n \mathbb{E}_{x \sim \mathcal{D}}[(\beta_{i,x} - \mathbb{E}_{\mathcal{D}}[\beta_{i,x}])(|h_i(x) - f(x)| - L_{\mathcal{D}}(h_i))]
\end{aligned}
$$

where the third equality is based on $\mathbb{E}[XY] = \mathbb{E}[X]\mathbb{E}[Y] + \mathbb{E}[(X - \mathbb{E}[X])(Y - \mathbb{E}[Y])]$. By Hoeffding's inequality, for any fixed $i$, we have $\mathbb{E}_{x \sim \mathcal{D}}[\beta_{i,x}] \leq \frac{\sum_{x \in U} \beta_{i,x}}{T} + \sqrt{\frac{2\log\frac{2}{\delta}}{T}}$ with probability at least $1 - \delta$ over the random choice of samples. Taking the union bound over all $i \in N$, we have with probability at least $1 - n\delta$:

$$\sum_{i=1}^n \mathbb{E}_{x \sim \mathcal{D}}[\beta_{i,x}] \leq \sum_{i=1}^n \left(\frac{\sum_{x \in U} \beta_{i,x}}{T} + \sqrt{\frac{2\log\frac{2}{\delta}}{T}}\right)$$

Finally, with probability at least $1 - n\delta$, we have:

$$L_{\mathcal{D}}\left(h_N\right) \leq \sum_{i=1}^{n}\left(\frac{\sum_{x \in U} \beta_{i,x}}{T}+\sqrt{\frac{2 \log \frac{2}{\delta}}{T}}\right) L_{\mathcal{D}}\left(h_i\right)+\Sigma_N$$

$$=\sum_{i=1}^{n}\left(\mathcal{V}_i+\sqrt{\frac{2 \log \frac{2}{\delta}}{T}}\right) L_{\mathcal{D}}\left(h_i\right)+\Sigma_N$$

where $\Sigma_N = \sum_i^n \mathbb{E}_{x \sim \mathcal{D}}[(\beta_{i,x}-\mathbb{E}_{\mathcal{D}}[\beta_{i,x}])(|h_i(x)-f(x)|-L_{\mathcal{D}}(h_i))]$. Thus, we prove Proposition 1.

## C.2 PROOF OF THEOREM 1

We will show $u_i = \phi_i(\mathcal{V}+\mathcal{V}^p)$ as the Shapley value of the combined game of $\mathcal{V}$ and $\mathcal{V}^p$, where we simply use the shorthand $\phi_i$ to represent $\phi_i(\mathcal{V})$.

Recall that

$$\mathcal{V}^p(C)=\left(\sum_{i \in C} \phi_i\right) \times \gamma, \ \forall C \subseteq N \text{ where } \gamma=\sum_{j \in N} p_j/(\mathcal{V}_N-\phi_j) .$$

Following the original setting in the main text of Theorem 1 where $\phi_i = \phi_i(\mathcal{V})$ and $r_i = \phi_i$, we have:

$$
\begin{aligned}
u_i &= \phi_i(\mathcal{V})+\phi_i(\mathcal{V}^p) \quad \text{(linearity of Shapley)} \\
&= \phi_i + \mathcal{V}^p(\{i\}) \quad \text{(additivity of } \mathcal{V}^p) \\
&= r_i + \sum_{j \in N} \frac{\phi_i \times p_j}{\mathcal{V}_N-\phi_j} \\
&= r_i + \frac{\phi_i \times p_i}{\mathcal{V}_N-\phi_i} + \sum_{j \in N \backslash\{i\}} \frac{\phi_i \times p_j}{\mathcal{V}_N-\phi_j} \\
&= r_i + \frac{\phi_i \times p_i + \mathcal{V}_N \times p_i - \mathcal{V}_N \times p_i}{\mathcal{V}_N-\phi_i} + \sum_{j \in N \backslash\{i\}} \frac{\phi_i \times p_j}{\mathcal{V}_N-\phi_j} \\
&= r_i + \frac{\mathcal{V}_N \times p_i - p_i \times (\mathcal{V}_N-\phi_i)}{\mathcal{V}_N-\phi_i} + \sum_{j \in N \backslash\{i\}} \frac{\phi_i \times p_j}{\mathcal{V}_N-\phi_j} \\
&= r_i + \underbrace{\frac{\mathcal{V}_N \times p_i}{\mathcal{V}_N-\phi_i}}_{r_i^+} + \underbrace{\sum_{j \in N \backslash\{i\}} \frac{\phi_i \times p_j}{\mathcal{V}_N-\phi_j}}_{p_i^+} -p_i
\end{aligned}
$$

$$(3)$$

which proves the utility $u_i$ as the Shapley value of the combined game, and it specifies the fair reward and payoff allocation.

From the above results, we also have another representation:

$$u_i = \left(1+\sum_{j \in N} \frac{p_j}{\mathcal{V}_N-\phi_j}\right) \times \phi_i . \tag{4}$$

### C.2.1 Proof of Payoff Balance Property

$$\sum_{i \in N}(p_i^+ - p_i)$$

$$= \sum_{i \in N}\left(\sum_{j \in N \setminus i}\frac{\phi_i \times p_j}{\mathcal{V}_N - \phi_j} - p_i\right)$$

$$= \sum_{i \in N}\sum_{j \in N \setminus i}\frac{\phi_i \times p_j}{\mathcal{V}_N - \phi_j} - \sum_{i \in N}p_i$$

$$= \sum_{i \in N}\frac{\left(\sum_{j \in N \setminus i}\phi_j\right) \times p_i}{\mathcal{V}_N - \phi_i} - \sum_{i \in N}p_i$$

$$= \sum_{i \in N}\frac{(\mathcal{V}_N - \phi_i) \times p_i}{\mathcal{V}_N - \phi_i} - \sum_{i \in N}p_i$$

$$= \sum_{i \in N}p_i - \sum_{i \in N}p_i$$

$$= 0.$$

### C.2.2 Proof of Dummy Payment Property

Since $\forall C \subseteq N \setminus \{i\}$, $\mathcal{V}(C \cup i) = \mathcal{V}(C)$, we have $\phi_i = 0$ by the definition of Shapley value. As $r_i^+ = (\mathcal{V}_N \times p_i)/(\mathcal{V}_N - \phi_i)$, we have $r_i^+ = (\mathcal{V}_N \times p_i)/\mathcal{V}_N = p_i$. Besides,

$$u_i = r_i + r_i^+ + \sum_{j \in N \setminus \{i\}}\frac{\phi_i \times p_j}{\mathcal{V}_N - \phi_j} - p_i$$

$$(\text{By } T1, r_i = k_1\phi_i = 0) \quad = 0 + p_i + \sum_{j \in N \setminus \{i\}}\frac{0 \times p_j}{\mathcal{V}_N - \phi_j} - p_i$$

$$= p_i - p_i$$

$$= 0$$

### C.2.3 Proof of Semi-Symmetry Property

Since $\forall C \subseteq N \setminus \{i, j\}$, $\mathcal{V}(C \cup i) = \mathcal{V}(C \cup j)$, we have $\phi_i = \phi_j$ by the definition of Shapley value. By Equation 4, we have

$$u_i = \left(1 + \sum_{j \in N}\frac{p_j}{\mathcal{V}_N - \phi_j}\right) \times \phi_i = \left(1 + \sum_{j \in N}\frac{p_j}{\mathcal{V}_N - \phi_j}\right) \times \phi_j = u_j.$$

### C.2.4 Proof of Strict Monotonicity Property

When $\exists j \in N \; p_j' > p_j$ and $\forall k \in N \; p_k' \geq p_k$, we have

$$(\text{By Equation 4}) \quad u_i = \left(1 + \sum_{j \in N}\frac{p_j}{\mathcal{V}_N - \phi_j}\right) \times \phi_i$$

$$= \left(1 + \frac{p_j}{\mathcal{V}_N - \phi_j} + \sum_{k \in N \setminus \{j\}}\frac{p_k}{\mathcal{V}_N - \phi_k}\right) \times \phi_i$$

$$< \left(1 + \frac{p_j'}{\mathcal{V}_N - \phi_j} + \sum_{k \in N \setminus \{j\}}\frac{p_k'}{\mathcal{V}_N - \phi_k}\right) \times \phi_i$$

$$= u_i'.$$

### C.3    PROOF OF PROPOSITION 2

We first introduce the definition of the divergence measure $d_{\mathcal{H}\Delta\mathcal{H}}$.

**Definition C.1** ($\mathcal{H}\Delta\mathcal{H}$-divergence (Ben-David et al., 2010))**.** *Let $h : \mathcal{X} \to \{0, 1\}$ be a function from the hypothesis class $\mathcal{H}$. $\mathcal{H}$-divergence between $\mathcal{D}_i$ and $\mathcal{D}_j$ is:*

$$d_{\mathcal{H}\Delta\mathcal{H}}(\mathcal{D}_i, \mathcal{D}_j) := 2 \sup_{h,h' \in \mathcal{H}} \left| L_{\mathcal{D}_i}(h, h') - L_{\mathcal{D}_j}(h, h') \right|$$

By the definition of $\mathcal{H}\Delta\mathcal{H}$-divergence,

$$d_{\mathcal{H}\Delta\mathcal{H}}(\mathcal{D}_i, \mathcal{D}_j) \geq 2 \left| L_{\mathcal{D}_i}(h, h') - L_{\mathcal{D}_j}(h, h') \right|. \tag{5}$$

With the use of triangle inequality for classification error (Crammer et al., 2008), for any labeling functions $f_1$, $f_2$, and $f_3$, we have $L(f_1, f_2) \leq L(f_1, f_3) + L(f_2, f_3)$. Note that the combined empirical risk is defined as $\hat{L}_{\mathcal{D}'_i}(h) := (1 - \alpha_i)\hat{L}_{\mathcal{D}_i}(h) + \alpha_i \hat{L}_{\mathcal{D}}(h, h_N)$ for each party $i \in N$. For any $h$, we have:

$$
\begin{aligned}
& |L_{\mathcal{D}'_i}(h) - L_{\mathcal{D}}(h)| \\
&= |(1 - \alpha_i)L_{\mathcal{D}_i}(h) + \alpha_i L_{\mathcal{D}}(h, h_N) - L_{\mathcal{D}}(h)| \\
&= |(1 - \alpha_i)[L_{\mathcal{D}_i}(h) - L_{\mathcal{D}}(h)] + \alpha_i[L_{\mathcal{D}}(h, h_N) - L_{\mathcal{D}}(h)]| \\
&\leq (1 - \alpha_i)|L_{\mathcal{D}_i}(h) - L_{\mathcal{D}}(h)| + \alpha_i|L_{\mathcal{D}}(h, h_N) - L_{\mathcal{D}}(h)| \\
&\leq \frac{1}{2}(1 - \alpha_i)d_{\mathcal{H}\Delta\mathcal{H}}(\mathcal{D}_i, \mathcal{D}) + \alpha_i|L_{\mathcal{D}}(h, h_N) - L_{\mathcal{D}}(h, f) + L_{\mathcal{D}}(h, f) - L_{\mathcal{D}}(h)| && \text{(E. 5)} \\
&\leq \frac{1}{2}(1 - \alpha_i)d_{\mathcal{H}\Delta\mathcal{H}}(\mathcal{D}_i, \mathcal{D}) + \alpha_i\left[|L_{\mathcal{D}}(h, h_N) - L_{\mathcal{D}}(h, f)| + |L_{\mathcal{D}}(h, f) - L_{\mathcal{D}}(h)|\right] \\
&\leq \frac{1}{2}(1 - \alpha_i)d_{\mathcal{H}\Delta\mathcal{H}}(\mathcal{D}_i, \mathcal{D}) + \alpha_i L_{\mathcal{D}}(h_N) && \text{(Triangle inequality)} \tag{6}
\end{aligned}
$$

where $f$ is the true labeling function, $L_{\mathcal{D}}(h, f) = L_{\mathcal{D}}(h)$, and $L_{\mathcal{D}}(h_N, f) = L_{\mathcal{D}}(h_N)$.

We can bound the combined risk with its empirical estimation through Hoeffding's inequality, which we state here:

**Lemma 1.** *(Ben-David et al., 2010) For a fixed hypothesis $h$, if there are $T_i$ random samples from domain $\langle \mathcal{D}, h_N \rangle$ and $m_i$ random samples from domain $\langle \mathcal{D}_i, f \rangle$, then for any $\delta \in (0, 1)$ and $t > 0$, with probability at least $1 - \delta$ (over the choice of the samples),*

$$\mathbb{P}\left( \left| L_{\mathcal{D}'_i}(h) - \hat{L}_{\mathcal{D}'_i}(h) \right| \geq t \right) \leq 2 \exp\left( \frac{-2t^2}{\frac{\alpha_i^2}{T_i} + \frac{(1 - \alpha_i)^2}{m_i}} \right).$$

Now we are ready to prove Proposition 2.

*Proof.* The proof is similar to the standard proof of uniform convergence for empirical risk minimizers. For the hypothesis $h'_i$ that minimizes the combined loss $\hat{L}_{\mathcal{D}'_i}(h) := (1 - \alpha_i)\hat{L}_{\mathcal{D}_i}(h) + \alpha_i \hat{L}_{\mathcal{D}}(h, h_N)$, we first show here its proof of its generalization bound. With probability at least $1 - \delta$, the generalization inequality holds. The first and the last inequality are direct applications of the Inequality 6. The second and the fourth inequality are based on Lemma 1, and also rely on sample symmetrization and

bounding the growth function by the VC dimension (Anthony et al., 1999; Ben-David et al., 2010).

$$L_{\mathcal{D}}(h'_i) \leq L_{\mathcal{D}'_i}(h'_i) + \frac{1}{2}(1-\alpha_i)d_{\mathcal{H}\Delta\mathcal{H}}(\mathcal{D}_i, \mathcal{D}) + \alpha_i L_{\mathcal{D}}(h_N) \quad \text{(Inequality 6)}$$

$$\leq \hat{L}_{\mathcal{D}'_i}(h'_i) + 2\sqrt{2d\log(2(T_i+m_i+1)) + 2\log(\frac{8}{\delta})}\sqrt{\frac{\alpha_i^2}{T_i} + \frac{(1-\alpha_i)^2}{m_i}}$$

$$+ \frac{1}{2}(1-\alpha_i)d_{\mathcal{H}\Delta\mathcal{H}}(\mathcal{D}_i, \mathcal{D}) + \alpha_i L_{\mathcal{D}}(h_N) \quad \text{(Lemma 1)}$$

$$\leq \hat{L}_{\mathcal{D}'_i}(h_i) + 2\sqrt{2d\log(2(T_i+m_i+1)) + 2\log(\frac{8}{\delta})}\sqrt{\frac{\alpha_i^2}{T_i} + \frac{(1-\alpha_i)^2}{m_i}}$$

$$+ \frac{1}{2}(1-\alpha_i)d_{\mathcal{H}\Delta\mathcal{H}}(\mathcal{D}_i, \mathcal{D}) + \alpha_i L_{\mathcal{D}}(h_N) \quad (h' := \arg\min_{h\in\mathcal{H}} \hat{L}_{\mathcal{D}'_i}(h))$$

$$\leq L_{\mathcal{D}'_i}(h_i) + 4\sqrt{2d\log(2(T_i+m_i+1)) + 2\log(\frac{8}{\delta})}\sqrt{\frac{\alpha_i^2}{T_i} + \frac{(1-\alpha_i)^2}{m_i}}$$

$$+ \frac{1}{2}(1-\alpha_i)d_{\mathcal{H}\Delta\mathcal{H}}(\mathcal{D}_i, \mathcal{D}) + \alpha_i L_{\mathcal{D}}(h_N) \quad \text{(Lemma 1)}$$

$$\leq L_{\mathcal{D}}(h_i) + 4\sqrt{2d\log(2(T_i+m_i+1)) + 2\log(\frac{8}{\delta})}\sqrt{\frac{\alpha_i^2}{T_i} + \frac{(1-\alpha_i)^2}{m_i}}$$

$$+ (1-\alpha_i)d_{\mathcal{H}\Delta\mathcal{H}}(\mathcal{D}_i, \mathcal{D}) + 2\alpha_i L_{\mathcal{D}}(h_N) \quad \text{(Inequality 6)}.$$

Therefore, we have
$$L_{\mathcal{D}}(h'_i) - \epsilon_i \leq L_{\mathcal{D}}(h_i)$$
where $\epsilon_i = 4\sqrt{2d\log(2(T_i+m_i+1)) + 2\log(\frac{8}{\delta})}\sqrt{\frac{\alpha_i^2}{T_i} + \frac{(1-\alpha_i)^2}{m_i}} + (1-\alpha_i)d_{\mathcal{H}\Delta\mathcal{H}}(\mathcal{D}_i, \mathcal{D}) + 2\alpha_i L_{\mathcal{D}}(h_N)$. Taking the union bound over all $i \in N$, we have probability at least $1 - n\delta$:

$$L_{\mathcal{D}}(h'_i) - \epsilon_i \leq L_{\mathcal{D}}(h_i), \ \forall i \in N.$$

Define $\epsilon = \max_{i\in N} \epsilon_i$. Then we have
$$\forall i \in N \ L_{\mathcal{D}}(h'_i) - \epsilon \leq L_{\mathcal{D}}(h_i).$$

This concludes the proof of Proposition 2.

### C.4 PROOF OF EQUATION 2

Recall that $\epsilon_i(\alpha) = B\sqrt{\alpha^2/T + (1-\alpha)^2/m} + \alpha A + d_{\mathcal{H}\Delta\mathcal{H}}(\mathcal{D}_i, \mathcal{D})$, where $A = -d_{\mathcal{H}\Delta\mathcal{H}}(\mathcal{D}_i, \mathcal{D}) + 2L_{\mathcal{D}}(h_N)$ and $B = 4\sqrt{2d\log(2(T_i+m_i+1)) + 2\log(\frac{8}{\delta})}$. Let $C = A^2/B^2$.

To find the optimal mixing value $\alpha^*$ that minimizes $\epsilon_i(\alpha)$, we take the derivative of $\epsilon_i(\alpha)$ over $\alpha$:

$$\frac{\partial \epsilon_i(\alpha)}{\partial \alpha} = \frac{1}{2}B\frac{1}{\sqrt{\frac{\alpha^2}{T} + \frac{(1-\alpha)^2}{m}}}\left(\frac{2\alpha}{T} + \frac{2(\alpha-1)}{m}\right) + A$$

$$= \frac{B\left(\frac{\alpha}{T} + \frac{\alpha-1}{m}\right)}{\sqrt{\frac{\alpha^2}{T} + \frac{(1-\alpha)^2}{m}}} + A.$$

By setting $\frac{\partial \epsilon_i(\alpha)}{\partial \alpha} = 0$, we have

$$\frac{\left(\frac{\alpha}{T} + \frac{1-\alpha}{m}\right)}{\sqrt{\frac{\alpha^2}{T} + \frac{(1-\alpha)^2}{m}}} = -\frac{A}{B}.$$

Squaring both sides results in

$$\frac{m^2\alpha^2 + 2Tm\alpha(\alpha - 1) + T^2(\alpha - 1)^2}{Tm^2\alpha^2 + T^2m(\alpha - 1)^2} = \frac{A^2}{B^2}.$$

By setting $C = A^2/B^2$ and rearranging, we have

$$[(m + T)^2 - CTm(m + T)]\alpha^2 + 2T(CTm - m - T)\alpha + T^2 - CT^2m = 0. \tag{7}$$

The discriminant of this quadratic equation of $\alpha$ is $\Delta = 4CT^2m^2(m + T - CTm)$. The equation has real roots when $\Delta \geq 0$, which is $m + T - CTm \geq 0 \Rightarrow m \geq (Cm - 1)T$.

When $Cm \leq 1$, no matter how large $T$ is, $\Delta \geq 0$ always holds, as $m$ and $T$ are positive integers.

When $Cm > 1$, $\Delta \geq 0 \Rightarrow \frac{m}{Cm-1} \geq T$. As the training data size $m$ is generally large enough such that $Cm > 1$ always holds, we will not analyse the case when $Cm \leq 1$.

Therefore, the above quadratic equation has real roots when $\frac{m}{Cm-1} \geq T$, with the corresponding quadratic solution given as:

$$\alpha^* = \frac{T}{m + T}\left(1 + \frac{m\sqrt{C}}{\sqrt{m + T - CTm}}\right).$$

It is not difficult to see that

$$\frac{\partial^2 \epsilon_i(\alpha)}{\partial \alpha^2} = \frac{B}{Tm\left(\frac{\alpha^2}{T} + \frac{(1-\alpha)^2}{m}\right)^{3/2}} \geq 0 .$$

Thus, the quadratic solution for $\frac{\partial \epsilon_i(\alpha)}{\partial \alpha} = 0$ is the minimum for $\epsilon_i(\alpha)$.

When $\frac{m}{Cm-1} < T$, Equation 7 has no real roots. As $\frac{\partial \epsilon_i(\alpha)}{\partial \alpha} < 0$, the minimum of $\epsilon_i(\alpha)$ is found at the boundary when $\alpha = 1$. Finally, checking with the boundary [0,1] of $\alpha$ results in the exact solution of $\alpha^*$ which is:

$$\alpha^* = \left\{ \begin{array}{ll} 1 & \text{for} \quad T > m(Cm - 1)^{-1} \\ \min\{1, \xi\} & \text{for} \quad T \leq m(Cm - 1)^{-1} \end{array} \right. \text{where} \; \xi = \frac{T}{m + T}\left(1 - \frac{m\sqrt{C}}{\sqrt{m + T - CTm}}\right).$$

This concludes our proof.

### C.5 PROOF OF EFFICIENT CALCULATION OF MODEL SHAPLEY VALUE

$$\begin{aligned}
\phi_i(\mathcal{V}) &= (1/n!) \sum_{\pi \in \Pi_N} [\mathcal{V}(C_{\pi,i} \cup \{i\}) - \mathcal{V}(C_{\pi,i})] \\
&= (1/n!) \sum_{\pi \in \Pi_N} [\sum_{x \in U} \sum_{j \in C_{\pi,i} \cup \{i\}} \beta_{j,x}/T - \sum_{x \in U} \sum_{j \in C_{\pi,i}} \beta_{j,x}/T] \\
&= (1/n!) \sum_{\pi \in \Pi_N} \left[\frac{1}{T} \sum_{x \in U} \left(\sum_{j \in C_{\pi,i} \cup \{i\}} \beta_{j,x} - \sum_{j \in C_{\pi,i}} \beta_{j,x}\right)\right] \\
&= (1/n!) \sum_{\pi \in \Pi_N} \left[\frac{1}{T} \sum_{x \in U} \beta_i\right] \\
&= \frac{1}{T} \sum_{x \in U} \beta_i
\end{aligned}$$

which concludes the proof.

## C.6 PROOF OF INCENTIVE GUARANTEE OF OUR ALLOCATION SCHEME

We prove below that our allocation scheme satisfies incentives T1 to T3:

• **T1.** Note we use $r_i = (\phi_i/\phi^*) \times \mathcal{V}_N$ in our allocation scheme. By our definition of utility, for all $i \in N$, we have

$$r_i/\phi_i = \mathcal{V}_N/\phi^* = k_1$$
$$\frac{r_i^+ - p_i + p_i^+}{\phi_i} = \frac{\mathcal{V}_N \times p_i - (\mathcal{V}_N - \phi_i) \times p_i}{(\mathcal{V}_N - \phi_i) \times \phi_i} + \frac{p_i^+}{\phi_i}$$
$$= \frac{p_i}{\mathcal{V}_N - \phi_i} + \sum_{j \in N \setminus \{i\}} \frac{p_j}{\mathcal{V}_N - \phi_j}$$
$$= \sum_{j \in N} \frac{p_j}{\mathcal{V}_N - \phi_j}$$
$$= k_2.$$

• **T2.** Refer to Sec. 5.2 for details of $L_\mathcal{D}(h_i') - \epsilon \leq L_\mathcal{D}(h_i)$ guarantee. For all $i \in N$, we have

$$r_i^+ - p_i = \frac{\mathcal{V}_N \times p_i - (\mathcal{V}_N - \phi_i) \times p_i}{\mathcal{V}_N - \phi_i}$$
$$= \frac{\phi_i \times p_i}{\mathcal{V}_N - \phi_i}$$
$$\geq 0$$

where $\phi_i$, $p_i$, and $\mathcal{V}_N - \phi_i \geq 0$.

• **T3.** There exists a party $i$ with $\phi_i = \phi^*$, such that

$$r_i = (\phi_i/\phi^*) \times \mathcal{V}_N = (\phi^*/\phi^*) \times \mathcal{V}_N = \mathcal{V}_N,$$
$$p_i^* = (\phi^* - \phi_i)(\mathcal{V}_N - \phi_i)/\phi^* = (\phi^* - \phi^*)(\mathcal{V}_N - \phi^*)/\phi^* = 0.$$

# D ADDITIONAL EXPERIMENTAL RESULTS AND DETAILS

All of our experiments are performed on a machine with AMD EPYC 7543 32-Core Processor, 256GB of RAM, and NVIDIA RTX 3080 GPU with 10GB memory.

## D.1 VALUATION

To justify our valuation method under the heterogeneous data setting, we follow the same partition scheme in (Lin et al., 2020) to synthesize some non-i.i.d source data distributions. Assume every party's training samples are drawn independently with class labels following a categorical distribution over $K$ classes. We draw $\tau \sim \text{Dir}(\tau \mathbf{p})$ from a Dirichlet distribution, where $\mathbf{p}$ characterizes a prior class distribution over $K$ classes, and $\tau > 0$ is a concentration parameter controlling the identicalness among local parties. We use $\tau = 0.1$ in our experiments. A neural network with two fully connected layers and ReLU activation functions is employed by each party to fit their respective training data. In Table. 3, we observe a strong negative correlation between $\mathcal{V}_i$ and $L_\mathcal{D}(h_i)$ on three datasets on 5 and 10 parties accordingly. The average ensemble weight $\mathcal{V}_i$ is calculated on a validation dataset of size 5000 using the optimal ensemble method.

(Fig. 6 and the following paragraph are added during rebuttal.)

We emphasize that the requirement on the ground truth of our method is not very restrictive, where the labeled data (i.e., ground truth) of size as small as 100 is sufficient to identify high-quality models, as shown in Fig. 6. It is shown that the correlation is stronger and the standard deviation is smaller when the size of labeled data becomes larger. Especially, we observe that the negative correlation of size 100 is almost as good as that of size 4000. We will also demonstrate the effectiveness of our

Table 3: Pearson correlation between $\mathcal{V}_i$ and $L_{\mathcal{D}}(h_i)$ on different datasets with different numbers of parties under a heterogeneous data setting. The value is reported with the mean and standard error over 100 independent evaluations.

|  | No. Parties | |
| --- | --- | --- |
| Dataset | 5 | 10 |
| MNIST | -0.72±0.35 | -0.72±0.35 |
| CIFAR-10 | -0.90±0.06 | -0.90±0.06 |
| SVHN | -0.83±0.11 | -0.83±0.11 |

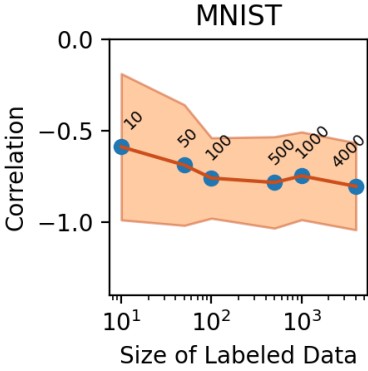

Figure 6: Pearson correlation between $\mathcal{V}_i$ and $L_{\mathcal{D}}(h_i)$ on MNIST with different ground truth size. The scatter points represent different ground truth size of [10, 50, 100, 500, 1000, 4000] respectively.

method using several practical ensemble methods later. In practice, those weights estimated on the small labeled dataset can be potentially used for other unlabeled data.

We also validate our valuation function on MNIST dataset under a heterogeneous model setting. The training data is divided into five random subsets using a symmetric Dirichlet distribution. We use five fully connected models to fit each subset of the data, with hidden layer sizes {}, {1024}, {512, 256}, {1024, 256}, and {1024, 512, 256} respectively and the ReLU activation functions. The output layer is a softmax layer that consists of 10 neurons. The parameters of the network are trained with the Adam optimizer. The learning rate is set to 0.0001, batch size is 128, and number of epochs is 5. We examine the Pearson correlation between $\mathcal{V}_i$ and $L_{\mathcal{D}}(h_i)$ over 100 independent evaluations, which gives a strong negative correlation coefficient of -0.76±0.21. It justifies our analysis in Proposition 1 again. The average ensemble weight is calculated on a validation dataset of size 5000 using the optimal ensemble method. Besides, we observe the average Shapley vector for the five models is [0.11, 0.23, 0.23, 0.22, 0.21], which shows models with more parameters might overfit the training data and may have a lower contribution in the collaboration.

As introduced in App. B.2, we further study how other different ensemble methods, instead of the ideal optimal ensemble method, affect the average ensemble weight $\mathcal{V}_i$ and the test accuracy using our model valuation function. We perform experiments on MNIST with 5 parties under both i.i.d. and non-i.i.d. data setting ($\tau = 0.1$). A neural network with two fully connected layers and ReLU activation functions is employed by each party to fit their respective training data. The average ensemble weight $\mathcal{V}_i$ is calculated on a validation dataset of size 5000 using different ensemble methods, and the accuracy of ensemble predictions is measured based on the labels given for the validation dataset. From Table. 4, we observe that the AVG method achieves comparable results with other ensemble methods on the accuracy of ensemble predictions, but it has a weak correlation coefficient, which shows that it is not suitable for contribution evaluation in collaboration. It is interesting to note that MV has a slightly stronger correlation reported in Table. 5 but achieves a lower accuracy. This seemingly counter-intuitive result can be attributed to the fact that its discrete weight of MV may not capture the relation between $\mathcal{V}_i$ and $L_{\mathcal{D}}(h_i)$ when irrelevant parties contribute to the ensemble prediction. We encourage further research to develop more sophisticated ensemble methods that can effectively capture the value of individual models by approximating the optimal ensemble method.

Table 4: Pearson correlation between $\mathcal{V}_i$ and $L_{\mathcal{D}}(h_i)$, and the corresponding accuracy of ensemble predictions with different ensemble methods under the i.i.d. data setting. The value is reported with the mean and standard error over 100 independent evaluations.

| Ensemble | Correlation | Accuracy |
|----------|-------------|----------|
| AVG | 0.02±0.54 | 0.95±0.01 |
| MV | -0.24±0.52 | 0.94±0.01 |
| KV | -0.25±0.46 | 0.95±0.01 |
| MWU | -0.24±0.56 | 0.95±0.01 |

Table 5: Pearson correlation between $\mathcal{V}_i$ and $L_{\mathcal{D}}(h_i)$, and the corresponding accuracy of ensemble predictions with different ensemble methods under the non-i.i.d. data setting. The value is reported with the mean and standard error over 100 independent evaluations.

| Ensemble | Correlation | Accuracy |
|----------|-------------|----------|
| AVG | -0.02±0.50 | 0.72±0.09 |
| MV | -0.65±0.32 | 0.68±0.08 |
| KV | -0.40±0.49 | 0.80±0.05 |
| MWU | -0.68±0.32 | 0.72±0.09 |

## D.2 GUARANTEE $\epsilon$-IR

We perform additional experiments to illustrate how ensemble predictions are fairly distributed as rewards to improve the model performance of individual parties, where the number of all ensemble predictions (i.e., $T$ can vary). We also show that the strict IR can be empirically satisfied and $\epsilon$-IR can have the strongest guarantee. In this section, we only consider the optimal ensemble method. We have $L_{\mathcal{D}}(h_N) = 0$ for the optimal ensemble. Through estimating $d_{\mathcal{H}\Delta\mathcal{H}}(\mathcal{D}_i, \mathcal{D})$ and the VC dimension $d$ based on Ben-David et al. (2010), we could find the optimal value $\alpha^*$ whose formulation is shown in Eqn. 2. The $d_{\mathcal{H}\Delta\mathcal{H}}(\mathcal{D}_i, \mathcal{D})$ is estimated as $2 \times (1 - 2\text{err})$ where err is the loss from a classifier that tries to separate the unlabeled data from $\mathcal{D}_i$ and $\mathcal{D}$. VC dimension $d$ is estimated as $c \cdot W \cdot L$ where $c$ is some constant, $W$ is the parameter size of the model, and $L$ is the number of layers.

### D.2.1 MNIST

MNIST dataset (LeCun et al., 1998) consists of 70000 handwritten digits (0-9), each of which is a $28 \times 28$ grayscale image. There are 60000 training images and 10000 test images. We follow the same experimental setting as (Nguyen et al., 2022) to partition the training dataset based on the class labels. A party, denoted as $[s - e]$ $(s \leq e)$, owns a subset of the training dataset that is labeled with classes $s, s + 1, \ldots, e$. We consider 5 parties in $N$: [0], [1-2], [0-3], [3-5], and [6-9]. Each party uses a deep neural network to fit its training images and only shares the predictions for an unlabeled dataset of size $T$ sampled from the training images.

A neural network with two fully connected layers and ReLU activation functions is employed by each party to fit their respective training data. The parameters of the network are trained with the Adam optimizer. The learning rate is set to 0.001, batch size is 128, and number of epochs is 20.

**Different** $T$. We examine the experimental results of improved model performance with rewarded ensemble predictions for varying values of $T$, specifically $T = 1000, 5000, 10000$. In this experimental setting, the optimal ensemble method is used. As shown in Fig. 7a&b, the model performance of all considered parties exhibits a notable improvement with an increase in the magnitude of $T$. Furthermore, it is interesting to observe that fair reward distribution could lead to equality in model performance when $T$ attains a sufficiently large value, as shown in Fig. 7c.

(Experiments on accuracy gain with different ensembles and Fig. 8 are newly added during the rebuttal, as shown in the following paragraph.)

**Different Ensembles**. In addition, we also present an examination of how various practical (non-optimal) ensemble methods influence the enhancement in accuracy after the rewards are fairly distributed. This experiment is conducted under the setting, where $T = 5000$ for the unlabeled dataset $U$. The 5 parties share their predictions of $U$ with the host, and the host uses some ensemble methods to produce the ensemble predictions, which are then distributed to each party according to our allocation scheme. For simplicity, we ignore the payment here. In Fig. 8, we demonstrate the accuracy improvement for each party by studying different practical ensemble methods, including AVG, KV, MWU, and MV. We observe that the strict IR (i.e., $L_{\mathcal{D}}(h_i') \leq L_{\mathcal{D}}(h_i)$) is always satisfied with all ensemble methods. For the result with MV in Fig. 8d, we can see that party [6-9] did not obtain a better performance after collaboration (i.e., $L_{\mathcal{D}}(h_i') = L_{\mathcal{D}}(h_i)$). Since party [6-9]'s data is

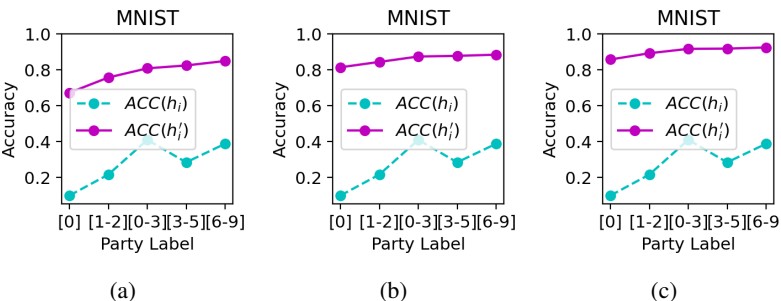

(a)          (b)          (c)

Figure 7: Plot of test accuracy of five parties' models on MNIST before and after incorporating fair ensemble prediction reward with different $T$: (a) $T = 1000$ (b) $T = 5000$, and (c) $T = 10000$.

unique, it cannot contribute to the ensemble predictions by the definition of MV. Therefore, it receives no reward from the collaboration. This suggests we should use appropriate ensemble methods such as KV or MWU in collaboration.

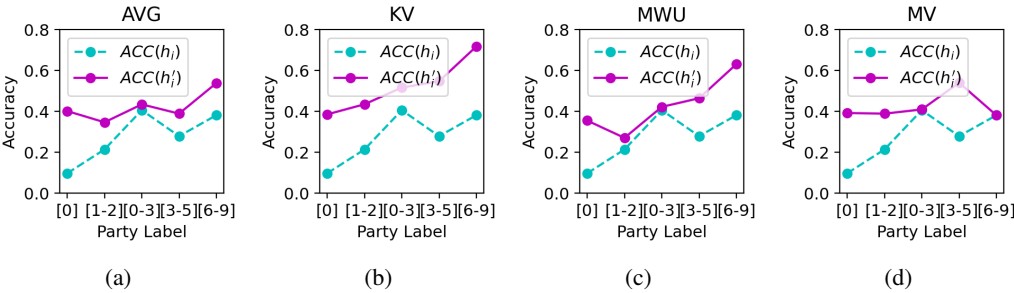

(a)          (b)          (c)          (d)

Figure 8: Plot of test accuracy of five parties' models on MNIST before and after incorporating fair ensemble prediction reward with different practical ensemble methods: (a) Average (AVG), (b) Knowledge Vote (KV), (c) Multiplicative Weight Update (MWU), and (d) Majority Vote (MV).

==(The following paragraph and Fig. 9 are added during the rebuttal, where we empirically show how to quantify the relation between payment and accuracy increase.)==

As the payment $p_i$ affects the number of ensemble predictions $T_i$ that party $i$ receives, and hence affects the accuracy gain, we will demonstrate the relation between payment and accuracy increase. If we use $L_{\mathcal{D}}(h_i) + \epsilon_i$ as an approximation of $L_{\mathcal{D}}(h_i')$, the change of $\epsilon_i$ denoted as $\Delta\epsilon_i$ will represent the reduction in error, which empirically reflects the accuracy increase. To quantify the relation between payment and accuracy increase, we can write $\Delta\epsilon_i$ as a function of $p_i$ shown below:

$$\Delta\epsilon_i(p_i) = M - 4\sqrt{2d\log(2(r_iT + \frac{p_iT}{1-\phi_i} + m_i + 1)) + 2\log(\frac{8}{\delta})}\sqrt{\frac{\alpha_i^2}{r_iT + p_iT/(1-\phi_i)} + \frac{(1-\alpha_i)^2}{m_i}}$$

where $M = 4\sqrt{2d\log(2(r_iT + m_i + 1)) + 2\log(\frac{8}{\delta})}\sqrt{\frac{\alpha_i^2}{r_iT} + \frac{(1-\alpha_i)^2}{m_i}}$ is a constant. When $p_i = 0$, $\Delta\epsilon_i(p_i) = 0$. When $p_i$ is larger, $\Delta\epsilon_i(p_i)$ will be larger. To empirically demonstrate the relation, we follow the previous five parties' collaboration example and examine the effect on party [0-3]'s payment. From Fig. 9, we observe that more payments lead to a higher accuracy increase, and $\Delta\epsilon_i(p_i)$ can indeed empirically represent the accuracy increase.

### D.2.2   CIFAR-10

We experiment on CIFAR-10 dataset (Krizhevsky et al., 2009), which consists of 60000 32×32 color images in 10 classes, with 6000 images per class. There are 50000 training images and 10000 test images in total. We follow the same experimental setting as (Nguyen et al., 2022) to partition the training dataset based on the class labels. A party, denoted as $[s - e]$ ($s \le e$), owns a subset of the training dataset that is labeled with classes $s, s + 1, \ldots, e$. We consider 5 parties in $N$: [0], [1-2],

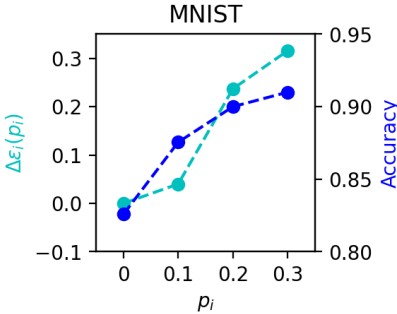

Figure 9: Plot of $\Delta\epsilon_i(p_i)$ and accuracy when party [0-3] makes different payment $p_i$.

[0-3], [3-5], and [6-9]. Each party uses a deep neural network to fit its training images and only shares the predictions for an unlabeled dataset of size $T$ sampled from the training images.

The neural network has 2 convolutional layers with number of filters 6 and 16. The kernel size of the two layers is set to (5,5). There are 2 max-pooling layers with size (2,2) after the 2 convolutional layers. The output from the last convolutional layers is flattened and passed to 2 hidden linear layers with the number of neurons 120 and 84, and the ReLU activation functions. The output layer is a softmax layer that consists of 10 neurons. The parameters of the network are trained with the Adam optimizer. The learning rate is set to 0.0001, batch size is 128, and number of epochs is 20.

We compare the experimental results when $T = 1000, 5000, 10000$, as shown in Fig. 10. At $T = 1000$, every participating party experiences an improvement in model performance. Interestingly, party [0-3], with relatively lesser contribution, achieves higher test accuracy than party [3-5] due to more source data. However, when $T$ increases to 5000 or 10000, party [3-5] attains superior test accuracy due to the larger rewards it receives, which correspond to its more significant contribution. This would promote more valuable parties like party [3-5] to collaborate.

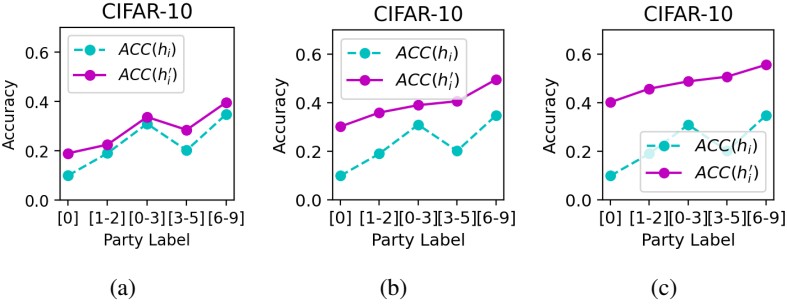

|     |     |     |
| :-: | :-: | :-: |
| (a) | (b) | (c) |

Figure 10: Plot of test accuracy of five parties' models on CIFAR-10 before and after incorporating fair ensemble prediction reward with different $T$: (a) $T = 1000$ (b) $T = 5000$, and (c) $T = 10000$.

We continue with the above experimental setting of CIFAR-10 when $T = 5000$. We consider both party [0-3] and party [6-9] as examples. Fig. 11a shows that our estimated $\alpha_i^*$ helps the party achieve the lowest generalization error which means the best model performance. The optimal $\alpha_i^*$ will always minimize $\epsilon_i(\alpha_i)$ as shown in Fig. 11b, which again validates the correctness of our solution in Eqn. 2. We can observe from Fig. 11c that $\alpha_i$ affects the generalization error drop $L_{\mathcal{D}}(h_i') - L_{\mathcal{D}}(h_i)$, and our optimal $\alpha_i^*$ results in the largest drop, which implies the greatest model improvement. Therefore, the strongest $\epsilon$-IR guarantee is achieved by the optimal $\alpha_i^*$. As we should have $L_{\mathcal{D}}(h_i') - L_{\mathcal{D}}(h_i) - \epsilon_i \leq 0$ from $\epsilon$-IR, we observe from Fig. 11d that $(L_{\mathcal{D}}(h_i') - L_{\mathcal{D}}(h_i) - \epsilon_i)$ is always negative with different $\alpha_i$. This means $\epsilon$-IR is always satisfied. Similar analysis can be applied to party [6-9]; however, there is a slight mismatch where $\alpha_i^*$ does not correspond to the least $L_{\mathcal{D}}(h_i')$ in Fig. 12a, which means the party might not achieve its best performance by using $\alpha_i^*$. Nevertheless, $\alpha_i^*$ is still the minimum of $\epsilon(\alpha_i)$, and $\epsilon$-IR is still satisfied as shown in Fig. 12c&d. This mismatch might be from the generalization error gap between $L_{\mathcal{D}}(h_i')$ and $L_{\mathcal{D}}(h_i)$.

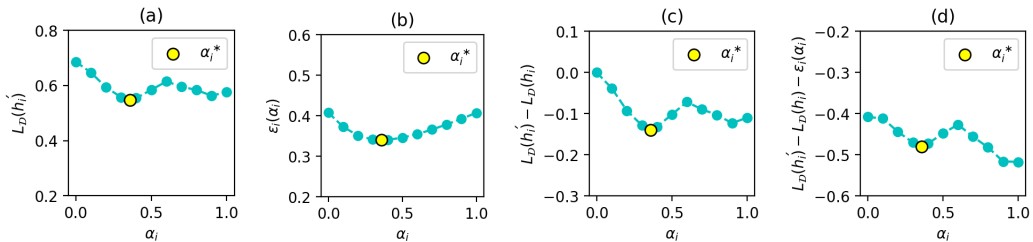

Figure 11: (a) The generalization error of $h'_i$ with different $\alpha_i$, (b) the function value of $\epsilon_i(\alpha_i)$ and its minimum $\epsilon_i(\alpha_i^*)$, (c) the generalization error drop $L_\mathcal{D}(h'_i) - L_\mathcal{D}(h_i)$ with different $\alpha_i$, and (d) the $\epsilon$-IR guarantee quantified by $L_\mathcal{D}(h'_i) - L_\mathcal{D}(h_i) - \epsilon_i(\alpha_i)$ with different $\alpha_i$, when party $i$ is party [0-3] in CIFAR-10.

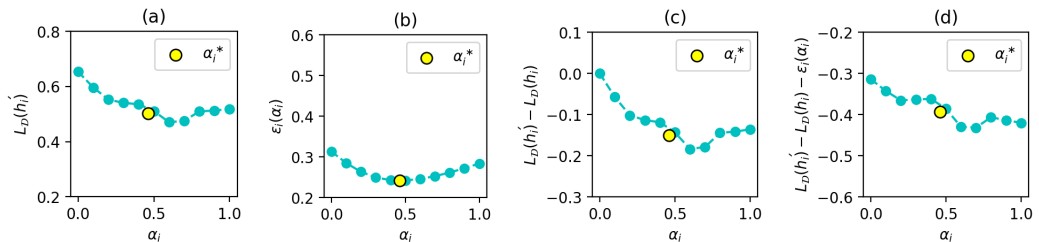

Figure 12: (a) The generalization error of $h'_i$ with different $\alpha_i$, (b) The function value of $\epsilon_i(\alpha_i)$ and its minimum $\epsilon_i(\alpha_i^*)$, (c) the generalization error drop $L_\mathcal{D}(h'_i) - L_\mathcal{D}(h_i)$ with different $\alpha_i$, and (d) the $\epsilon$-IR guarantee quantified by $L_\mathcal{D}(h'_i) - L_\mathcal{D}(h_i) - \epsilon_i(\alpha_i)$ with different $\alpha_i$, when party $i$ is party [6-9] in CIFAR-10.

### D.3 FAIR UTILITY

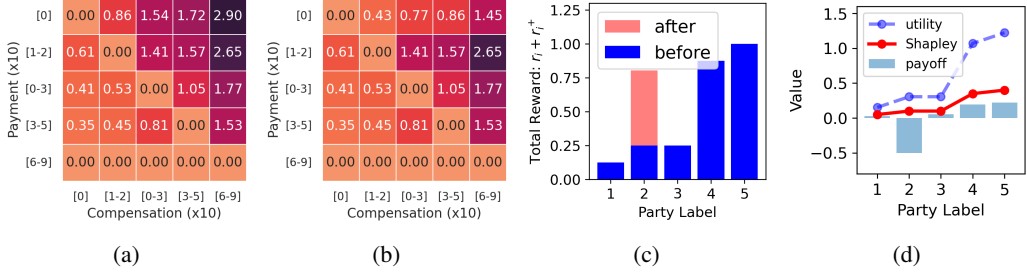

Figure 13: (a) The payoff flow when all parties make their maximal payments, (b) the payoff flow when party [0] makes less payment while others make their maximal payments, (c) the rewards that the parties receive before and after the payment made by party 2, and (d) the utility, Shapley value and utility of the parties when party 2 makes a payment.

We adopt the experimental setting of partitioned MNIST dataset where each party owns a subset of digits: [0], [1-2], [0-3], [3-5], and [6-9]. We calculate the corresponding maximal payments $(\phi^* - \phi_i)(\mathcal{V}_N - \phi_i)/\phi^*$ for the five parties: [0.702, 0.624, 0.376, 0.315, 0.0]. Fig. 13a shows the payoff flow when parties make their maximal payments, where each entry shows the payment from row party to column party. Parties with larger contributions (e.g., party [6-9]) benefit more from each payment made by another party. Next, we assume there is a budget constraint of party [0] limiting it to make half its maximal payment. The corresponding payoff flow in Fig. 13b shows the change of payment from one party will not affect the payoff flow of other parties.

We then examine the semi-symmetry property of our utility function. We consider a hypothetical collaboration for 5 parties (i.e., $N = \{1, ..., 5\}$), where the corresponding Shapley value vector is

[0.05, 0.1, 0.1, 0.35, 0.4] and only party 2 makes a payment of 0.5. Fig. 13c shows the reward of party 1 increases after it makes the payment with the rewards of others remaining the same. From Fig. 13d, we observe the semi-symmetry property, where party 2 and 3 have the same Shapley and utility values, but different payoffs. It ensures fairness in collaboration when parties with same contributions but different financial budgets.

