# OpenReview forum: "Incentivized Black-Box Model Sharing"
_ICLR.cc/2024/Conference — Submitted to ICLR 2024_

### Official Review · Reviewer_jHme · 2023-10-27

**Soundness:** 3 good
**Presentation:** 3 good
**Contribution:** 3 good
**Rating:** 8
**Confidence:** 1

**Summary:**

In this paper, the authors introduced an incentivized black-box model sharing framework that equitably distributes ensemble predictions and rewards parties based on their contributions. The authors (1) introduced a Weighted Ensemble Game to quantify the contribution of black-box models towards predictions; (2) derived a closed-form solution for fair reward allocation based on Weighted Ensemble Game and  Fair Replication Game; (3) theoretically proved that approximate individual rationality is satisfied. Finally, the authors also conduct numerical experiments on real-world data to confirm the efficacy of their method.

**Strengths:**

Overall, this paper is well written and clearly addresses the three main questions that the authors proposed to address, each corresponding to (1) how to quantify the contributions made by each model, (2) how to ensure that each party receives a fair payment/reward and (3) how to ensure individual rationality is ensured. It also provides solid theoretical results for each of the aforementioned questions, accompanied by empirical evaluations.

Nonetheless, I am not an expert in the field of Black-Box Model Sharing and hence have limited expertise in evaluating the merit/weakness of this work.

**Weaknesses:**

See questions.

**Questions:**

(1) Could you provide one specific example that motivates why individual rationality is chosen as one of your key metrics?

(2) Why do you consider Shapley fairness as your main fairness notion? Any other fairness notions that might fit into your framework?

(3) In Sec 5 you suggested that "We will later empirically show that the virtual regret $\epsilon$ is not needed and the strict IR is satisfied". Is this a purely empirical observation or do you believe stronger theoretical results can be established here?

---

> ### Author Response · Authors · 2023-11-16
>
> We thank Reviewer jHme for reviewing our paper and for the positive and encouraging feedback!
>
> We hope to answer your questions as follows.
> ***
> Q1:
> > Could you provide one specific example that motivates why individual rationality is chosen as one of your key metrics?
>
> Let's consider a scenario where three private hospitals (A, B, and C) want to share predictions based on their patient data to improve the predictive performance of their model. However, Hospital C has already invested significant resources into collecting a high-quality dataset (with a high data diversity), while Hospitals A and B have datasets that are noisy or of lower quality in comparison. If the ensemble predictions are naively shared among all three hospitals, Hospitals A and B would benefit more because they can significantly improve their models, while it may not benefit Hospital C at all (i.e., **no collaborative gain** in the form of either monetary or improved model) and its competitors are better off. In other words, hospital C did not improve its utility by collaborating (i.e., the individual rationality is *not* satisfied) and would be disincentivized to collaborate. Therefore, enforcing individual rationality can ensure Hospital C's collaboration by ensuring its utility will improve (or positive collaborative gain).
>
> Additionally, a similar example is used to motivate individual rationality in the case of autonomous driving to achieve a win-win outcome (in the form of legislation that enforces individual rationality) for both consumers and car companies [1].
>
> Therefore, individual rationality is one key incentive that we need to consider in collaboration.
>
> [1] Karimireddy, S. P., Guo, W., & Jordan, M. I. (2022). Mechanisms that incentivize data sharing in federated learning. *arXiv preprint* arXiv:2207.04557.
>
> ***
>
> Q2:
> > Why do you consider Shapley fairness as your main fairness notion?
>
> Intuitively, the more one contributes, the more it should receive from collaboration. To formalize this intuition, there are many ways to do so. In our problem, we adopt the Shapley value as an interesting and meaningful way to achieve this because it is the unique solution that satisfies the four fairness properties (*efficiency*, *symmetry*, *dummy party*, and *linearity*) as in App. B.1.
>
> - [**symmetry**] To ensure two equally contributing parties are equally recognized, symmetry ensures that two *parties with equal marginal contributions to any coalition in the collaboration receive the same collaborative gain*;
>
> - [**dummy party**] To prevent *free-riders*, dummy party ensures that *parties with zero marginal contributions to any coalition receive no collaborative gain*.
>
>     Additionally, it is shown by Sim et al. (2020) that Shapley value also satisfies some other fairness properties: strict desirability and strict monotonicity, which we have newly added to App. B.1. Hence, the Shapley value is a careful **design choice** in our work.
>
> As additional justification, the Shapley value is a very widely used solution in previous data sharing works (Ghorbani & Zou, 2019; Jia et al., 2019a; Sim et al., 2020; Tay et al., 2022; Karimireddy et al., 2022;) as the notion of fairness.
>
> > Any other fairness notions that might fit into your framework?
>
> One other common fairness notion, known as *egalitarian*, aims at achieving equitable outcomes for all the parties (e.g., by rewarding all the parties equally). This does not satisfy the dummy party property and can cause the free-rider problem since even the non-contributing party is rewarded equally with the high-contributing parties.
>
> Another common fairness notion (in the allocation of goods) is *envy-freeness*, which is not applicable here. Because it is typically used for rivalrous goods, but the rewards here (i.e., ensemble predictions) are non-rivalrous (since they can be replicated), and it is the reason for our proposed definition of fair replication game.
>
> For discussions of different fairness notions, we will incorporate them in the revision.
>
> ***
> We answer the other question in the subsequent comment.

---

> ### Author Response · Authors · 2023-11-16
>
> Q3:
> >In Sec 5 you suggested that "We will later empirically show that the virtual regret $\epsilon$ is not needed and the strict IR is satisfied". Is this a purely empirical observation or do you believe stronger theoretical results can be established here?
>
> It is *mainly an empirical observation*, but we think *stronger theoretical results may be possible*. The reason is the derivations for our theoretical result on the virtual regret $\epsilon$ involve several inequalities (e.g., Hoeffding's inequality, the triangle inequality, the constraint of the growth function by the VC dimension, and the supremum inequality of the domain divergence measure, as in App. C.3.). The improvement of the tightness of one or more of these inequalities can potentially lead to a stronger theoretical result. We will include this comment in our revision.
> ***
> We hope our responses have addressed your concerns and helped you to understand our work better. We are happy to answer any additional questions.

---

> > ### Author Response · Authors · 2023-11-22
> > **We would like to know if you have any further questions that require additional clarification**
> >
> > Dear Reviewer jHme,
> >
> > We sincerely appreciate the time and effort you've dedicated to reviewing our paper, as well as the insightful questions you've raised.
> >
> > If you have any more questions, we are eager to provide prompt responses.
> >
> > Thanks,
> >
> > Authors

---

### Official Review · Reviewer_hnks · 2023-10-31

**Soundness:** 2 fair
**Presentation:** 1 poor
**Contribution:** 3 good
**Rating:** 5
**Confidence:** 3

**Summary:**

The paper studies how to incentivize different agents to participate in black-box model sharing.

More specifically, given a set of points S, the host wants each agent to share their predictions on those points, and the host incentivizes them by giving the final ensemble predictions over these points (every agent's predictions are weighted by some weights beta), which can be used to get a new and hopefully improved model h'. The number of these additional points and the ensemble predictions on these points given to each agent is proportional to the contribution of the agent. They show a principled manner of how to measure contribution of each agent. Also, they show how to incentivize each agent to actually participate here: i.e. there's incentive for them to report their predictions because the new model h'trained with the addition of the points and ensemble predictions performs better than the previous model h.

Each agent can make a payment to collect more of those points and their ensemble predictions. And the paper shows how to set up these payment values and reward values so as to guarantee some form of fairness (T1 on pg 5).

They also evaluate their approach on some datasets.

**Strengths:**

-The main problem that they study is well-motivated, and the guarantees that they seek seem reasonable as well. It's nice that they can verify the theoretical claims in their experiments.

**Weaknesses:**

-My main complaint of the paper is that the overall presentation was pretty hard to follow, resulting in some confusion over few details of the paper.  For instance, I’m a little confused about how the weights beta_{i,x}’s are set if the true label for point x is unknown. See more detailed question below. And also, it seems that there’s an assumption about the unique of the optimal ensemble weights. Anyway, I think it would be helpful to add more prose to improve the overall presentation of the paper; I think the valuation part in section 6 is not too surprising but can be used as a sanity check and be moved to the appendix, which will allow more room to add more prose throughout the paper.

**Questions:**

-The paper describes once how the ensemble weights are set in 4.1. However, here it’s assumed the host actually knows the ground truth. So, is it just that in the very beginning where the host has access to a data set that’s held off, the host asks the clients to participate and find these weights in the very beginning and use these weights going forward?  But more realistically, the host would want to query each party to provide predictions for points for which the true label is unknown. In those cases, how would want find these weights? Note that the way things are written, the weight beta_{i,x} is set differently for each point x, meaning one can’t estimate these beta_{i,x} differently for each x, if the true label for that y is not known, but rather set a weight beta_i that’s the same across all the points. This should still maintain proposition 1, as all the arguments are always averaged over the entire distribution D anyway.


-I think there’s an inherent assumption that the optimal weights beta’s are unique. Consider a following example where every party has the same exact model h. Then, the ensemble model will be the same no matter how the weights beta’s are set.  In this case because everyone has the same model, one should be rewarded the same reward, meaning the beta’s should be uniform across every client. However, setting beta’s such that it places all its weight on a single model is also an optimal solution, which results in only that client receiving all the rewards. I think this is not just an artifact of this toy example, but if the data that each client has is pretty homogenous and resulting in similar overall loss, this can be very possible (assuming that as I described above the weights should be chosen not over (party i, point x) but rather over just the parties).

---

> ### Author Response · Authors · 2023-11-16
>
> We are grateful to the reviewers for their constructive and detailed feedback. We will incorporate this feedback into our revised work. We respond below to their concerns:
> ***
> W1:
> >I’m a little confused about how the weights $\beta_{i,x}$’s are set if the true label for point x is unknown.
>
> Please kindly refer to our response below in Q1 and **App. B.2** for how $\beta_{i,x}$’s are set if the true label for point x is unknown.
>
> >add more prose throughout the paper.
>
> To avoid significant changes (listed below) in the rebuttal as instructed by the guideline in ICLR 2024 "*Area chairs and reviewers reserve the right to ignore changes that are significantly different from the original paper*", we will make the following changes in the revision:
>
> (1) move the valuation part in section 6 to Appendix
>
> (2) move some remarks in Appendix to the main paper to add more elaborations
>
> (3) include more experiments with non-optimal (practical) ensemble methods in the main paper
>
>
> ***
> Q1:
> >So, is it just that in the very beginning ... the host asks the clients to participate and find these weights in the very beginning and use these weights going forward?
>
> No. In our setting, no matter what ensemble method is used, the host will query each party for each data point and continually determine the ensemble weights.
>
> >But more realistically, the host would want to query each party to provide predictions for points for which the true label is unknown.
>
> Your realistic consideration is correct. In fact, __in some of our experiments__, the host queries each party for each data point and continually determines the ensemble weights, where the true label is unknown.
>
>
> >...for which the true label is unknown. In those cases, how would the host want find these weights?
>
> We assume the host knows the ground truth **only when** we use the optimal ensemble method. As we have described in **App. B.2**, ensemble methods from previous works such as average ensemble (AVG), majority vote (MV), knowledge vote (KV), and multiplicative weight update (MWU) do **not** need the ground truth. Those ensemble methods can set the weight $\beta_{i,x}$ both **differently** for each party and each data $x$.
>
> On the other hand, if the ground truth is available, we can observe a much stronger correlation (i.e., **-0.72**) between the average ensemble weight $\mathcal{V}\_i$ and the generalization error $L\_{\mathcal{D}}(h\_i)$ on MNIST in Table 2, compared with the correlation of MV (i.e., **-0.24**) and MWU (i.e., **-0.24**) on MNIST in table 1.
>
> We conduct an **additional experiment** (update in App. D.1) and highlight that the requirement on the ground truth is **not** very restrictive, where the labeled data (i.e., ground truth) of size as small as 100 is sufficient to identify high-quality models, as shown in Fig. 6 in App. D.1. The negative correlation of size 100 is as good as that of size 4000. In practice, those weights estimated on the small labeled dataset can be potentially used for other unlabeled data.
>
> ***
> We address the other concern in the subsequent comment.

---

> > ### Author Response · Authors · 2023-11-16
> >
> > Q2:
> > >Consider a following example where every party has the same exact model h ... In this case because everyone has the same model, one should be rewarded the same reward, meaning the beta’s should be uniform across every client.
> >
> > In our experiment, the beta’s are indeed **uniform** across every client when they have the **same** exact model $h$, because we set the initial value of $\beta_{i,x}^\ast=1/n$ in solving linear optimization problem of the optimal ensemble. This implementation detail can be found in our provided code (in line 75 in the file "server/collaboration.py").
> >
> > >I think this is not just an artifact of this toy example, but if the data that each client has is pretty homogenous and resulting in similar overall loss, this can be very possible.
> >
> > Your understanding is correct. Translating your intuition of "pretty homogenous" data for $i\neq i'\in N$ into the following: if $h_i(x)$ and $h_{i'}(x)$ are "close", then we want $\beta_{i,x} \approx \beta_{i',x}$. To achieve this in implementation, we add the following regularization term $\sum_{\substack{i,i'\in N \\\ i\neq i'}} (|\beta_{i,x}-\beta_{i',x}|)^{(2-\||h_i(x)-h_{i'}(x)\||_1)}$ to the original linear optimization problem, as follows:
> >
> > $
> > \underset{\beta_{i,x}^*\in [0,1],\forall i \in N.}{\text{minimize}}\ |f(x)-{\textstyle\sum_{i=1}^n\beta_{i,x}^*h_i(x)}|+\lambda \sum_{\substack{i,i'\in N \\\ i\neq i'}} (|\beta_{i,x}-\beta_{i',x}|)^{(2-\||h_i(x)-h_{i'}(x)\||_1)}
> > $
> >
> > $\text{subject to}\ {\textstyle\sum_{i=1}^n\beta_{i,x}^*=1}.$
> >
> > The hyperparameter 2 in $(2-\||h_i(x)-h_{i'}(x)\||_1)$ is because the L-1 norm of the difference is upper bounded by 2. Please kindly refer to App. B.2 for the detailed **updated** formulation.
> >
> > Let's illustrate this concept with an example: Suppose we have a collaboration involving five parties. Here, each party shares a 3-class prediction vector $h_i(x)$ of the query data $x$. The ground truth for this data is $\mathbf{[1.0, 0.0, 0.0]}$. Refer to the table below for $h_i(x)$ values from each party. Specially, we impose that $h_1(x)$ and $h_{i\neq 1}(x)$ are "close".
> >
> > | Party  | $h_i(x)$ | $\beta_{i,x}^\ast$ w/o the regularization | $\beta_{i,x}^\ast$ w/ the regularization  |
> > |---|:---:|:---:|:---:|
> > | 1 | [1.0, 0.0, 0.0] | 1.0  | 0.2016  |
> > | 2 | [0.9, 0.1, 0.0] | 0.0  | 0.1996  |
> > | 3 | [0.9, 0.1, 0.0] | 0.0  | 0.1996  |
> > | 4 | [0.9, 0.1, 0.0] | 0.0  | 0.1996  |
> > | 5 | [0.9, 0.1, 0.0] | 0.0  | 0.1996  |
> >
> > We observe that $\beta_{1,x}^\ast$ of party $1$ is $1.0$ and $\beta_{i,x}^\ast$ of other parties are 0 when we **exclude** the regularization term, even though $h_i(x)$ and $h_{i'}(x)$ are "close". However, when we **include** the regularization term, we can then achieve $\beta_{i,x} \approx \beta_{i',x}$ for $i\neq i'\in N$ as desired.
> >
> >
> > ***
> > We hope our responses have addressed your concerns and improved your opinion about our work. We are happy to answer any additional questions.

---

> > > ### Author Response · Authors · 2023-11-22
> > > **We would like to know if you have any further questions that require additional clarification**
> > >
> > > Dear Reviewer hnks,
> > >
> > > Thank you for taking the time to review our paper and for your valuable questions.
> > >
> > > If you have any more questions or need more details, we are happy to answer them promptly within the discussion period.
> > >
> > > Best,
> > >
> > > Authors

---

> > > > ### Comment · Reviewer_hnks · 2023-11-22
> > > > **Thanks for the clarifications**
> > > >
> > > > Sorry about the late response to the rebuttal, and thanks for the clarifications!
> > > >
> > > > Since there isn't that much time remaining for the rebuttal period, I'll be sure to make my questions brief. I'm still a little hung up on the weights when the ground truth is not known.
> > > >
> > > >
> > > > Don't a lot of the theoretical results hinge upon the weights being chosen according to the optimal weights when the ground truths are known? For instance, I think intuitively that if I just choose the weights arbitrarily (e.g. just always listen to a particular agent for whatever reason), I shouldn't expect to get a lot of the nice properties describe in Section 5.
> > > >
> > > > It seems to me that the proposed approaches when the ground truth is not known are mostly 'heuristics' to be close to the optimal weights; I do see that for multiplicative weights, you can in fact get a no-regret style closeness to the optimal weights.
> > > >
> > > > It would be great if my concern here is valid or not.
> > > >
> > > >
> > > > And adding a regularization term to enforce a uniqueness of the optimal weights makes sense to me!

---

> > > > > ### Author Response · Authors · 2023-11-23
> > > > > **Thanks for the response**
> > > > >
> > > > > We thank Reviwer hnks for acknowledging our clairifications, and wish to address the questions as follows.
> > > > >
> > > > > ***
> > > > >
> > > > > >Don't a lot of the theoretical results hinge upon the weights being chosen according to the optimal weights when the ground truths are known?
> > > > >
> > > > > We wish to clarify that this is **not** the case. Our theoretical results in Sec. 5 do not specifically require the optimal weights or knowing the ground truth. Indeed, "*our allocation scheme does not depend on any particular ensemble method*" (see the last three lines in the paragraph below Remark 1). We will make this clear in our revision.
> > > > >
> > > > > >For instance, I think intuitively that if I just choose the weights arbitrarily ... I shouldn't expect to get a lot of the nice properties described in Section 5.
> > > > >
> > > > > The "nice properties" will **still hold**, but we believe that we should not extend our considerations to any arbitrary ensemble method.
> > > > >
> > > > > - The properties (i.e., Shapley fairness, IR, and Theorem 1) described in Sec. 5 **still hold** with *arbitrary weights* (i.e., $\beta_{i,x}$). The measured fair contribution $\phi_i = \sum_{x\in U}\beta_{i,x}/T$ in Sec. 4 only specifies the contribution based on a **given** ensemble method (not the optimal one), and our method in Sec. 5 is designed to satisfy these properties w.r.t.~$\phi_i$, without explicit dependence on the ensemble: As shown in the proofs in Apps. C.2 and C.6, our theoretical results do not require specific weights $\beta_{i,x}$.
> > > > >
> > > > > - Nevertheless, we believe that the choice of ensemble method should not be arbitrary, because (1) effective ensemble methods (not requiring knowing the ground truth) are available, and (2) a sub-optimal ensemble method leads to ineffective collaboration, as elaborated later.
> > > > >
> > > > > >It seems to me that the proposed approaches when the ground truth is not known are mostly 'heuristics'...
> > > > >
> > > > > We assume that by "proposed approaches", the reviewer refers to the specific ensemble methods (i.e., AVG, MV, and MWU), and we wish to clarify that these methods are existing methods and **not** our proposed solutions.
> > > > >
> > > > >
> > > > > >It would be great if my concern here is valid or not.
> > > > >
> > > > > Your concern can be addressed, by using an effective ensemble method even if the ground truth is **not** available.
> > > > >
> > > > > In addition to our discussion above (our theoretical results are not conditioned on the optimality of the ensemble method), our empirical observations demonstrate that some existing ensemble methods perform reasonably well (compared to the optimal ensemble) in App. D.2.1, without knowing the ground truth. Hence, we primarily consider effective ensemble methods instead of any arbitrary method, because:
> > > > >
> > > > >   1. Existing ensemble methods, such as KV and MWU, already perform reasonably well. We summarize the results from Fig. 7(b) and Fig. 8 (b,c) into the following table, which demonstrates that the quality/optimality of the ensemble (method) has an effect on the accuracy gains of the parties: Optimal ensemble leads to the highest accuracy gain (which is expected), the effective ensembles (i.e., MWU and KV) both perform reasonably. We highlight that our method can be adapted to other (more effective) ensembles designed in the future.
> > > > >
> > > > > | Party  | Original Acc | Improved Acc (MWU)  | Improved Acc (KV)  | Improved Acc (Optimal)  |
> > > > > |---|:---:|:---:|:---:|:---:|
> > > > > | [0] | 0.098 | 0.389  | 0.392  | 0.801  |
> > > > > | [1-2] | 0.215 | 0.240  | 0.433  | 0.842  |
> > > > > | [0-3] | 0.407 | 0.422  | 0.528  | 0.890  |
> > > > > | [3-5] | 0.280 | 0.457  | 0.556  | 0.912  |
> > > > > | [6-9] | 0.382 | 0.632  | 0.760  | 0.925  |
> > > > >
> > > > >   2. A pooly chosen sub-optimal ensemble method can lead to an ineffective collaboration of the parties, as an arbitrary ensemble or adversarial ensemble (see the example in the last paragraph in Sec. 4) can produce meaningless ensemble predictions (i.e., with very high ensemble error $L_{\mathcal{D}}(h_N)$) that will not benefit any party.
> > > > >
> > > > > >And adding a regularization term to enforce a uniqueness of the optimal weights makes sense to me!
> > > > >
> > > > > Thank you for confirming our proposed regularization design!
> > > > >
> > > > > We hope to have clarified your questions about when the ground truth is not known, and helped improve your opinion of our work.

---

> ### Comment · Reviewer_hnks · 2023-11-23
> **Thanks for the response**
>
> I'm aware that the results still hold true regardless of what the chosen weights are, but my point is that the results hold only with respect to the given weights.
>
> So, if I choose arbitrary weights, the contribution determined by these weights would still be arbitrary and the theoretical guarantees would be arbitrary as it's with respect to those arbitrary weights. Therefore, in order to fully capture the contribution of each agent faithfully, we better choose "good" weights that truly captures the contribution of each agent because the good theoretical guarantees hinge upon the fact that the weights are faithfully measuring the contribution. Simply said, the quality of the theoretical guarantees hinge upon the quality of the weights.
>
> And my concern is that coming up with good weights and verifying that those are good weights is hard to do when there are no ground-truth; the paper is only empirically showing the previous ensemble methods seem to do well, but these are just practical heuristics, and there's no theoretical guarantee that these are good weights. Note that even in the experiments, a hold-out dataset is being used to show that the ensemble methods are performing well. But in the absence of any hold-out dataset with ground truths, how can one even tell whichever chosen ensemble method is actually obtaining good weights or not?
>
> Or am I still misunderstanding the results?

---

> > ### Author Response · Authors · 2023-11-23
> > **Thanks for the question**
> >
> > We thank Reviewer hnks for the further questions and wish to provide the following clarifications.
> >
> > ---
> >
> > >coming up with good weights and verifying that those are good weights is hard to do when there are no ground-truth
> >
> > We acknowledge that finding and verifying the good weights without **any ground-truth** is hard.
> >
> > >the paper is only empirically showing the previous ensemble methods seem to do well, but these are just practical heuristics, ...
> >
> > Note that our objective is to design an incentive framework for black-box model sharing for a given ensemble method, because previous works did not consider the problem of valuation and incentive-aware allocation. Our objective is *not* to propose a new ensemble method.
> >
> > >there's no theoretical guarantee that these are good weights...But in the absence of any hold-out dataset with ground truths, how can one even tell whichever chosen ensemble method is actually obtaining good weights or not?
> >
> > To directly answer the question, it is **very challenging** to provide theoretically guaranteed **good weights** "*in the absence of any hold-out dataset with ground truths*"; fortunately, our empirical observation shows that a relatively small dataset (with ground truths) is sufficient.
> >
> > To elaborate, under the PAC-learning framework (as the theoretical guarantee) [1,2], there is a *lower bound* on the sample complexity of such a hold-out dataset:
> > - Any ($\epsilon,\delta$) PAC-learning algorithm needs a labeled dataset (i.e., ground truth) with size $\Omega(\max (n \log d, d \log n)/\epsilon)$ (Theorem 6 from [1]) where $n$ is the number parties and $d$ is the VC dimension of the hypothesis class.
> >
> > This lower bound demonstrates the necessity of the hold-out dataset with ground truth for theoretical guarantees.
> >
> > As a verification, we perform an estimation [3] on the VC dimension for the ML model (neural network) used for MNIST.
> >
> > - From [3], the VC dimension is estimated with $O(W L \log (W))$ where $W=7840$ is the number of weights and $L=1$ is the number of layers for the neural network we used for MNIST.
> >
> > We find that the lower bound above using the estimated VC dimension is around $100$x of the size of the dataset in Fig. 6 in App. D.1, which we attribute to the omitted constants in the big-Oh notations of both the sample complexity lower bound and that of the VC dimension bound in [3]. This empirical observation shows that, though a labeled dataset is *necessary* (for theoretical guarantees), a *relatively small size* can be sufficient to determine good weights.
> >
> > We hope our response has helped clarify your questions.
> >
> > **References**
> >
> > [1] Chen, J., Zhang, Q., & Zhou, Y. (2018). Tight bounds for collaborative PAC learning via multiplicative weights. *Proc. NeurIPS*.
> >
> > [2] Blum, A., Haghtalab, N., Procaccia, A. D., & Qiao, M. (2017). Collaborative PAC learning. *Proc. NeurIPS*.
> >
> > [3] Harvey, N., Liaw, C., \& Mehrabian, A. (2017). Nearly-tight VC-dimension bounds for piecewise linear neural networks. *Proc. COLT*.

---

### Official Review · Reviewer_gZai · 2023-10-31

**Soundness:** 3 good
**Presentation:** 2 fair
**Contribution:** 2 fair
**Rating:** 6
**Confidence:** 2

**Summary:**

* This paper proposes a theoretical framework for incentivized black-box model sharing, based on cooperative games.
* On the first stage of interaction, each party $i\\in[n]$ trains a multiclass classifier $h_i(x)$ using distribution $\\mathcal{D}_i$, but are interested in maximizing performance on a different distribution $\\mathcal{D}$.
* The trained classifiers are sent to a trusted party, and combined into an ensemble model $h_N(x)=\\sum_i \\beta_{i,x} h_i(x)$. The trusted party evaluates $h_N$ on a dataset $U\\sim\\mathcal{D}^T$ from the target distribution, and performance is translated into fair rewards $r_i$ for each party by the weighted ensemble game (WEG) mechanism.
* The WEG mechanism is based on Shapley values of a fully-additive cooperative game. The contribution of the $i$-th party is assumed to be equal to the average ensemble weight of their predictor ($\\sum_{x\\in U} \\beta_{i,x}/T$).
* On the second stage, each party is allowed to add $p_i$ monetary funds to increase their reward, and additional rewards $r_i^+$ and payments $p_i^+$ are distributed fairly by the fair replication game (FRG) mechanism, relying on Theorem 1.
* Once the final reward values are set, rewards ($r_i+r_i^+$) are realized as iid samples from the set $\\{(x,h_N(x)\\}_{x \\in U}$, and offset payments $p_i-p_i^+$ are realized as monetary transfers.
* Empirical evaluation is performed on MNIST, CIFAR-10 and SVHN, demonstrating accuracy gains in several settings.

**Strengths:**

* Problem is well-motivated. Two-stage collaborative game structure is an interesting design approach.
* Makes effort to support key assumptions (e.g for valuation functions).
* Empirical evaluation supports claims and provides confidence bounds. Documented code is provided.

**Weaknesses:**

* Limitations of the proposed method are not discussed clearly.
* Unclear applicability for practical ensemble methods: Average ensemble weight is uncorrelated with the objectives of the parties (Table 1), experiments are performed with an "ideal method" (Section 4.1).
* Presentation is dense, and was hard for me to follow. Many remarks which were very helpful to my understanding only appeared in Appendix A.

**Questions:**

* Motivation: Under which conditions is the model incentive structure realistic, and the valuation assumption applicable? In the hospital example mentioned in Appendix A (Q2), it is reasonable to assume that every hospital has access to a data source $\\mathcal{D}_i$ based on their local population, however it doesn’t seem intuitive to me that the hospital would desire a classifier that has good performance on a population $\\mathcal{D}$ which is different than their own, and common to all other hospitals. Can you clarify this example, or give a different practical example where assumptions intuitively hold?
* How does the method perform under practical (non-ideal) ensemble methods?
* Price of fairness: If I understand correctly, it seems that the overall welfare of the parties ($\\sum_i L_{\\mathcal{D}}(h_i)$) would be maximized by sharing all target-dataset data $\\{(x_t,h_N(x_t)\\}_{t=1}^T$ with all parties. What are the shortcomings of this approach? How does its welfare compare to the mechanism presented in the paper?
* What is the relation between the objective $L_\\mathcal{D}(h_i)$ and the utility $u_i$ presented in Theorem 1? Also, is it possible to quantify the relation between payment and accuracy increase for a given problem instance?
* Technical questions: What is the meaning of the notation $\\hat{L}_{\\mathcal{D}}(h,h_N)$ in Section 5.2? Is there an upper bound on the size of realized reward $T_i$?

---

> ### Author Response · Authors · 2023-11-16
>
> We thank Reviewer gZai for taking the time to review our paper and for providing a very detailed summary and questions and positive feedback that our problem is **well-motivated**, our design approach is **interesting**, and for appreciating our efforts in supporting key assumptions and empirical evaluations.
>
> We would like to address the comments as follows.
>
> ***
> W1:
> > Limitations of the proposed method are not discussed clearly.
>
> We have added a section (App. A.1) discussing the limitations and will make this clearer in our revision.
>
> For example, one limitation of our method is that it is developed for ensemble methods that take the form of weighted sum formulation (in Section 4.1), which does appear in several common ensemble methods (mentioned in Section 4.1).
>
>
> ***
> W2:
> > Unclear applicability for practical ensemble methods
>
> We indeed have experimental *results for non-ideal ensemble methods* (which are more practical) and have also *included additional experiments*.
>
> - [Results for non-ideal ensemble methods.] The results are in Tables 4 and 5 (in App. D.1). We observe a **stronger correlation** between the average ensemble weight $\mathcal{V}\_{i}$  and the generalization error $L\_{\mathcal{D}}(h\_{i})$ with the practical ensemble methods (i.e., MV, KV, and MWU) as shown in Table 5 under the non-i.i.d. data setting, which we believe is more important because the setting is more realistic. If the ensemble is bad (e.g., AVG shown in Table 5), our method of valuation and allocation still works, but parties with better models may not be identified as such in the collaboration.
>
> - [Additional experiments.] We included **additional results** in our paper, in Fig. 8 (in App. D.2.1), to demonstrate the accuracy gains w.r.t. different non-ideal (practical) ensemble methods used. In particular, we observe that *strict IR is always satisfied*.
>
>     For the additional experiment on CIFAR-10, we will include it if it is complete before the end of the rebuttal; otherwise, we will include it in the revision.
>
> As these different (non-ideal) ensemble methods produce predictions of varying qualities, we use the ideal (i.e., optimal ensemble) method to illustrate the effect of an ideal case of collaboration. The non-ideal ensemble methods might not be as effective in achieving fairness or IR as the ideal ensemble, but they can be applied nevertheless.
>
> ***
>
> W3:
> > Many remarks which were very helpful to my understanding only appeared in Appendix A.
>
> Thank you for taking the time to carefully read our prepared appendix. It would be very helpful if the reviewer would let us know the specific remarks (currently in appendix) that would aid the understanding of the reader, if moved to the main paper. We will improve the presentation accordingly.
>
> ***
> We address the other concerns in the subsequent comment.

---

> > ### Author Response · Authors · 2023-11-16
> >
> > Q1:
> > > Under which conditions is the model incentive structure realistic, and the valuation assumption applicable?
> >
> > The model incentive structure is realistic when the parties are **self-interested** (i.e., interested in accuracy gain and monetary compensation) and desire to perform well on the same target domain $\mathcal{D}$ (please refer to the first paragraph in Sec. 3 for the formal problem formulation).
> >
> > The valuation assumption is applicable when the ensemble method follows the **weighted sum formulation** of $h_N(x) = \sum_{i=1}^n\beta_{i,x}h_i(x)$ (in Sec. 4.1).
> >
> > >In the hospital example mentioned in Appendix A (Q2), it is reasonable to assume that every hospital has access to a data source $\mathcal{D}_i$ based on their local population, however it doesn’t seem intuitive to me that the hospital would desire a classifier that has good performance on a population $\mathcal{D}$ which is different than their own, and common to all other hospitals. Can you clarify this example, or give a different practical example where assumptions intuitively hold?
> >
> > Consider the condition of a pandemic like COVID-19, that has an impact globally, affecting all hospitals and individuals. The data of different hospitals may have acquisition bias due to the different demographics of the patients, so the hospitals' source data $\mathcal{D}_i$ over COVID-19 variants would be heterogeneous. Individuals may be affected by other COVID-19 variants from other distributions in the future. In such a situation, each hospital is interested in developing generalizable classifiers to cater to all individuals by facilitating collaboration and would be dealing with the same population $\mathcal{D}$ [1].
> >
> > Moreover, existing works on collaborative learning often adopt this same setting of learning from multiple domains to perform well on the same target domain (Chang et al., 2019; Lin et al., 2020; Feng et al., 2021).
> >
> > We can also extend to different target domains in future work (as mentioned in Sec. 7) to design a collaboration that enables customized improvements (w.r.t. each $\mathcal{D}_i$).
> >
> > [1] Peiffer-Smadja, N., Maatoug, R., Lescure, F. X., D’ortenzio, E., Pineau, J., & King, J. R. (2020). Machine learning for COVID-19 needs global collaboration and data-sharing. *Nature Machine Intelligence*, 2(6), 293-294.
> >
> > ***
> >
> > Q2:
> > >How does the method perform under practical (non-ideal) ensemble methods?
> >
> > Please see the response to W2 above.
> > ***
> > Q3:
> > >Price of fairness: If I understand correctly, it seems that the overall welfare of the parties $(\sum_{i\in N}L_{\mathcal{D}}(h_i))$ would be maximized by sharing all target-dataset data with all parties.
> >
> > Yes, your understanding is correct.
> >
> > > What are the shortcomings of this approach?
> >
> > It can cause the **free-rider problem**, if we simply maximize the overall welfare regardless of the contributions. To elaborate, if there is a party that contributes nothing to the collaboration and yet still receives all the ensemble predictions (as every other party). This is **unfair** to the parties that make meaningful contributions and can thus disincentivize other parties (Sim et al., 2020).
> >
> > > How does its welfare compare to the mechanism presented in the paper?
> >
> > Under the case where all the parties have *no budget constraints* and make the maximal payment $p_i^\ast$ (see in the 7th line in the paragraph below Remark 1), our proposed mechanism can also **maximize the overall welfare**. In this case, **fairness** is still achieved by distributing the payment fairly, as the less-contributing parties will compensate the top contributors via monetary payments.
> >
> > If there is any *budget constraint*, the maximum overall welfare may not be achieved. So, there is a price to maximize overall welfare. Which one to choose will depend on the problem setting.

---

> > > ### Author Response · Authors · 2023-11-16
> > >
> > > Q4:
> > > > What is the relation between the objective $L_{\mathcal{D}}(h_i)$ and the utility $u_i$ presented in Theorem 1?
> > >
> > > The utility $u_i$ is the value that quantifies the received reward $(r_i + r_i^+)$ and the payoff $(p_i^+ - p_i)$. The generalization error $L_{\mathcal{D}}(h_i)$ can be reduced with incorporating ensemble predictions (of size $T_i$) realized from $(r_i + r_i^+)$.
> > >
> > > From Proposition 2, $L_{\mathcal{D}}(h_i') \leq L_{\mathcal{D}}(h_i)+\epsilon_i$ and $\epsilon_i$ depends on $T_i:=(r_i+r_i^+)\times T$. The total reward $(r_i + r_i^+)$ in utility $u_i$ determines $T_i$, and affects the improved generalization error $L_{\mathcal{D}}(h_i')$. Specifically, if the payoff $p_i^+ - p_i$ is fixed, a larger $u_i$ could lead to smaller $L_{\mathcal{D}}(h_i')$.
> > >
> > > >Is it possible to quantify the relation between payment and accuracy increase for a given problem instance?
> > >
> > >
> > > Yes, and we present an *informal* idea: Note that any party $i$ receives ensemble predictions of size $T_i:=(r_i+r_i^+)\times T$. The payment $p_i$ is made to purchase the additional reward of value $r_i^+ = \frac{\mathcal{V}_N\times p_i}{\mathcal{V}_N - \phi_i}$, which is realized as additional samples of ensemble predictions of size $(r_i^+ \times T)$. Therefore, the larger payment $p_i$ that party $i$ makes, the more ensemble predictions it will receive.
> > >
> > > From Proposition 2, $L_{\mathcal{D}}(h_i') \leq L_{\mathcal{D}}(h_i)+\epsilon_i$ and $\epsilon_i$ depends on $T_i$, where the payment $p_i$ is implicitly embedded in $T_i$. Therefore, a higher payment means a larger $T_i$, hence smaller $\epsilon_i$. If we use $L_{\mathcal{D}}(h_i)+\epsilon_i$ as an approximation of $L_{\mathcal{D}}(h_i')$, $\Delta\epsilon_i$ can represent the reduction in error, which is empirically **reflected as the increase in accuracy**. To **quantify the relation between payment and accuracy increase**, we write $\Delta\epsilon_i$ as a function of $p_i$ shown below:
> > > \\[
> > > \Delta\epsilon_i(p_i) = M  - 4 \sqrt{2d\log(2(r_iT+\frac{p_iT}{1 - \phi_i}+m_i+1))+2\log(\frac{8}{\delta})}\sqrt{\frac{\alpha_i^2}{r_iT+p_iT/(1 - \phi_i)}+\frac{(1-\alpha_i)^2}{m_i}}
> > > \\]
> > >
> > > where
> > > $M=4 \sqrt{2d\log(2(r_iT+m_i+1))+2\log(\frac{8}{\delta})}\sqrt{\frac{\alpha_i^2}{r_iT}+\frac{(1-\alpha_i)^2}{m_i}}$ is a constant. When $p_i=0$, $\Delta\epsilon_i(p_i)=0$. When $p_i$ is larger, $\Delta\epsilon_i(p_i)$ will be larger, and thus a higher accuracy.
> > >
> > > This idea is empirically verified in an **additional empirical result** in Fig. 9 in App. D.2.1.
> > >
> > > ***
> > >
> > > Q5:
> > > >Technical questions: What is the meaning of the notation $\hat{L}_{\mathcal{D}}(h,h_N)$ in Section 5.2? Is there an upper bound on the size of realized reward $T_i$?
> > >
> > > $\hat{L}\_{\mathcal{D}}(h,h\_N)$ is the empirical error of training with the ensemble predictions. Denote $U\_i=\\{x\_t, h_N(x\_t)\\}\_{t=1}^{T\_i}$ the set of ensemble predictions that party $i$ receives. By definition, $\hat{L}\_{\mathcal{D}}(h,h\_N) = \frac{1}{T\_i}\sum\_{x \in U\_i}\ell(h(x),h\_N(x))$.
> > >
> > > There is an upper bound on the size of realized reward $T_i=(r_i+r_i^+)\times T$, where $T_i$ is always less or equal to the number of all ensemble predictions $T$. The size $T$ is specified by the size $|U|$ of the distillation dataset $U$.
> > >
> > > ***
> > >
> > > We thank the reviewer again for the detailed questions and constructive feedback and hope that our response has answered your questions and helped raise your opinion of our work. We are happy to provide further clarifications.

---

> > > > ### Comment · Reviewer_gZai · 2023-11-21
> > > >
> > > > Thank you for the detailed response, and for the clarifications made in the revision. I have no further questions.
> > > >
> > > > For improving clarity (W3), I found Figure 5 and Appendix A.2 Q2 to be helpful, and I believe that a greater emphasis on concise intuitive examples will improve the overall clarity of the exposition.

---

> > > > > ### Author Response · Authors · 2023-11-22
> > > > > **Thank you!**
> > > > >
> > > > > Dear Reviewer gZai,
> > > > >
> > > > > We thank you very much for reviewing our response. We are glad that our response has addressed all your concerns.
> > > > >
> > > > > We will improve the clarity of the exposition accordingly in the revision.
> > > > >
> > > > > Best Regards,
> > > > >
> > > > > Authors

---

### Official Review · Reviewer_1Wgz · 2023-11-01

**Soundness:** 3 good
**Presentation:** 2 fair
**Contribution:** 2 fair
**Rating:** 5
**Confidence:** 2

**Summary:**

This paper introduces a framework for model sharing across parties. In relation to prior work, this paper considers incentives, as well as parties only sharing their model (rather than data which can be sensitive). The framework distributes rewards in proportion to the contribution of each party, and also allows for payments between parties.

**Strengths:**

- Tackles an important and practical problem of considering incentives in the context of model sharing
- Model enforces desirable properties such as fairness and IR, and combines many practical considerations together
- Analysis is thorough

**Weaknesses:**

The main weakness is in the exposition - I was not able to understand the model. It seemed like the model and problem formulation were not comprehensively specified. The fact that there is an FAQ section on the model speaks to how the model is not completely clear. Here are my questions that I couldn’t find answers to:
- How should we compare prediction error to monetary payments to "rewards" (samples of ensemble predictions)? (Do they use the same unit of measurement?)
- Relatedly, what is the formula for the utility of party i?
- Payments can be made from one party to another. Does each party, decide on their own, how much to pay to each other party, or is this transfer also specified as part of the mechanism? Does each party have a budget?

The model has two main parts, as described in Figure 5. Can we simply de-couple these two stages and study each part separately, or are there interactions that require studying them together? Just studying one aspect would make the paper simpler and more clear.

**Questions:**

see above

---

> ### Author Response · Authors · 2023-11-16
>
> We thank Reviewer 1Wgz for taking the time to review our paper and for recognizing that our studied problem is **important and practical**, our model achieves **desirable properties**, and our analysis is **thorough**. We wish to provide the following clarifications and have incorporated some changes in our revision (with highlighted text in the updated pdf).
>
> ---
>
> >The fact that there is an FAQ section on the model speaks to how the model is not completely clear.
>
> Our provided Q\&A section in App. A.2 is meant to further supplement the main paper with more nuanced and specific details. In our revision, we will improve the exposition of our work to specify the problem formulation more clearly.
>
> W1:
> > How should we compare prediction error to monetary payments to "rewards" (samples of ensemble predictions)? (Do they use the same unit of measurement?)
>
> The monetary payments (i.e., $p_i$) and the value $r_i$ of the rewards (of size $T_i$) can be viewed to use the same unit of measurement (i.e., in the same domain). This is by design so that the allocation mechanism (Theorem 1) enables a translation between ensemble predictions (i.e., the realized rewards $T_i$) and monetary payment. We note that this has not been previously achieved by existing works.
> Please refer to Q4 in App. A where we illustrate how the monetary currency is projected into the positive real domain $\mathbb{R}_+$ to align with the reward domain.
>
> When you say "compare prediction error to monetary payments", we assume you mean the relationship between the two. To elaborate more on the relationship: An increase in the monetary payment $p_i$ leads to a higher reward $r_i$, and thus a larger size for $T_i$, which can decrease the upper bound on the prediction error $L_{\mathcal{D}}(h_i')$:
>
> - From Proposition 2, the upper bound of **prediction error** (i.e., $L_{\mathcal{D}}(h_i')$) depends on **payment** $p_i$, because $L_{\mathcal{D}}(h_i') \leq L_{\mathcal{D}}(h_i)+\epsilon_i$ and $\epsilon_i$ depends on $T_i$. To compare the prediction error to monetary payment with approximation, we investigate "the change in $\epsilon_i$" as a function of $p_i$ in a newly added empirical result (i.e., Fig. 9 in App. D.2.1), which provides an empirical quantification between the accuracy gain w.r.t. the increase in (monetary) payment.
> ***
> W2:
> >Relatedly, what is the formula for the utility of party i?
>
> By **definition**, $u_i$ is the Shapley value to the linearly combined game of $G$ and $G^p$. By **derivation**, the explicit **formula** for the utility $u_i$ of party $i$ is given in Theorem 1: $u_i = r_i + r_i^+ + p_i^+ - p_i $, where
> - $r_i=\phi_i$,
> - $r_i^+ = \frac{\mathcal{V}_N\times p_i}{\mathcal{V}_N - \phi_i}$,
> - $p_i^+=\sum_{j \in N \setminus i }\frac{\phi_i \times p_j}{\mathcal{V}_N - \phi_j}$, and
> - $p_i\in \mathbb{R}_+$ is determined by the party itself.
>
> The utility $u_i$ represents the *overall gain* of party $i$ from the collaboration, as a sum of $i$'s total reward $(r_i + r_i^+)$ and payoff/net compensation $(p_i^+-p_i)$.
>
> - $(r_i + r_i^+)$ represents the *value* of ensemble predictions that party $i$ receives from the collaboration,
> - $p_i^+$ represents the value of the received monetary compensation from all other parties, and
> - $p_i$ is the payment that party $i$ made to the host in exchange for additional rewards of the value $r_i^+$.
>
> Our formula of the utility fully specifies all the terms. Note that these interpretations (of the utility) are provided in the main paper (above Theorem 1 and below Remark 1).
>
> ---
> W3:
> > Payments can be made from one party to another. Does each party, decide on their own, how much to pay to each other party, or is this transfer also specified as part of the mechanism?
>
> Each party decides on its own how much to pay *in total* (i.e., $p_i$) to the central host; the exact distribution of $p_i$ to each other party is **specified by our mechanism** (and managed by the central host) to achieve fairness: The proposed formula of $p_i^+=\sum_{j\in N\setminus \{i\}}\frac{\phi_i \times p_j}{\mathcal{V}_N - \phi_j}$ in Theorem 1 specifies this payment distribution from every party $j\in N\setminus \{i\}$ to party $i$.
>
> > Does each party have a budget?
>
> Our mechanism can achieve fairness whether or not a party has a budget constraint:
>
> - [With budget constraint.] If party $i$ has a budget of $B_i$, it can freely choose payment on its own as long as $p_i \leq B_i$, and fairness is always achieved by the reward design and realization, as shown in App. C.6.
>
> - [W/o budget constraint.] If party $i$ does not have any budget constraint, then it can make the maximal payment $p_i^\ast = (\phi^\ast -\phi_i)(\mathcal{V}_N-\phi_i)/\phi^\ast$ to receive all the reward of value $\mathcal{V}_N$, which is mentioned in the paragraph below Remark 1. In this case, fairness is achieved by fairly distributing the payments.
>
> ---
> We address the other concerns in the subsequent comment.

---

> > ### Author Response · Authors · 2023-11-16
> >
> > W4:
> > > Can we simply de-couple these two stages and study each part separately, or are there interactions that require studying them together? Just studying one aspect would make the paper simpler and more clear.
> >
> > We consider these two stages together (1) so that we can study the **overall gain** (i.e., $u_i$) of a party $i$ from the entire collaboration; and (2) because the $G^p$ depends on the outcome of $G$:
> >
> > 1. Considering the two stages together enables us to study the **overall gain** (i.e., $u_i$) of a party received from collaboration including the *reward* (i.e., $r_i + r_i^+$) and *payoff* (i.e., $p_i^+ - p_i$). Note that, in the paragraph above Theorem 1, we formally couple the two stages via two games, namely $G$ and $G^p$. The outcome of $G$ determines $r_i$, and the outcome of $G^p$ determines $r_i^+ - p_i + p_i^+$. With this coupling, we can directly determine the number for ensemble predictions $T_i=(r_i+r_i^+)\times T$ that party $i$ receives.
> >
> > 2. The fair replication game $G^p$ shown in Definition 5.1 depends on the outcome (i.e., $\\{\phi_i\\}_{i=1}^n$) from $G$, because the fairness in the monetary gain (i.e., $r_i^+ - p_i + p_i^+$) relies on the contribution measure $\phi_i$. To satisfy the Shapley fairness incentive (i.e., fairly allocating rewards and payoffs), it is necessary to calculate the Shapley value from the first stage. Hence, the two stages need to be coupled.
> >
> > Let's consider a concrete scenario of private hospitals collaborating by sharing predictions of unlabeled medical data. Small hospitals with lower contributions (i.e., a model of lower quality/performance) can make payments to purchase more rewards (i.e., ensemble predictions) to improve their models. Hence, it is important for the payment (in exchange for rewards) to be carefully based on the outcome of the first stage so that these parties (i.e., small hospitals) are able to obtain a high utility (i.e., improvement in their local models' performance).
> >
> > We will make these reasons (for considering the two stages) more explicit in our revision.
> >
> > ***
> > We hope our clarifications have addressed your questions and helped improve your opinion of our work. We are happy to answer any additional questions.

---

> > > ### Author Response · Authors · 2023-11-22
> > > **We would like to know if you have any further questions that require additional clarification**
> > >
> > > Dear Reviewer 1Wgz,
> > >
> > > Thank you for taking the time to review our paper and for your valuable questions.
> > >
> > > If you have any more questions or need more details, we are happy to answer them promptly within the discussion period.
> > >
> > > Best,
> > >
> > > Authors

---

> > > > ### Comment · Reviewer_1Wgz · 2023-11-22
> > > >
> > > > Thank you to the authors for their detailed clarifications. My original questions have been clarified.
> > > >
> > > > I still think that the paper is very dense and difficult to follow. Specifically, section 3 should completely specify the model, which it currently does not do rigorously. I found reviewer gZai's summary of the paper to be a clearer description of the model. Here are a couple of examples of sources of confusion:
> > > > - The reward is not defined - I believe it corresponds to a scalar value, but it is initially introduced as a set of predictions.
> > > > - This section should also clearly delineate which aspects of the process will be specified as the main contributions of the paper (Section 4+5). For example, after the sentence "parties are allowed to make monetary payments $p_i$ ..." - the authors should write that this payment mechanism will be detailed in Section 5 (and perhaps the desirable properties of this mechanism should also be written here). Essentially, it was unclear which parts are taken as given / as definition, and which parts represent the main contribution.
> > > > - There is a paragraph about the valuation function, but at this point it is completely unclear why this is relevant and how this relates to the model specified in the previous paragraph. It is written that the shapley value represents the "fair contribution of party i", but the fair contribution was never defined.
> > > >
> > > > I don't need the authors to respond individually to the above points since this is late (my apologies). But in summary, I found the paper difficult to follow because there was no roadmap and I felt that many descriptions and terms came out of nowhere. Looking at the paper, I do think much of my concerns can be alleviated if Section 3 was significantly clarified.

---

> > > > > ### Author Response · Authors · 2023-11-23
> > > > > **Thanks for the response**
> > > > >
> > > > > We thank Reviewer 1Wgz for acknowledging that our clarifications have **clarified the questions**. We wish to provide the following response:
> > > > >
> > > > > ***
> > > > > >The reward is not defined - I believe it corresponds to a scalar value, but it is initially introduced as a set of predictions.
> > > > >
> > > > > We define the value of reward in the second paragraph in Sec. 5 (i.e., "let $r_i\in \mathbb{R}_+$ denote the numerical value of party $i$'s reward ..."):
> > > > >
> > > > > - The *realized reward* for party $i$ is a set $U_i$ of data subsampled from the ensemble predictions, and its size is $T_i$ i.e., $|U_i|=T_i$ (see Q5 in App. A2).
> > > > > - The *reward value* is $(r_i+r_i^+) \in \mathbb{R}$, and $T_i = (r_i+r_i^+) \times T$ (see **Reward Realization** in Sec. 5.1).
> > > > >
> > > > > >This section should also clearly delineate which aspects of the process will be specified as the main contributions of the paper.
> > > > >
> > > > > We have incorporated your suggestion in our **updated** Sec. 3 in the **revised paper** (see pdf).
> > > > >
> > > > > >it was unclear which parts are taken as given / as definition, and which parts represent the main contribution.
> > > > >
> > > > > In our updated pdf, Sec. 3 now presents an overview of the proposed mechanism with the illustrative diagram (Fig. 1) previously in App. A, in which we clearly describe the definitions and contributions. Also, we note that our specific contributions are summarized in the last paragraph in Sec. 1.
> > > > >
> > > > > >...the valuation function, but at this point it is completely unclear why this is relevant...
> > > > >
> > > > > The valuation function $\mathcal{V}$ is designed so that we can utilize the Shapley value to fairly quantify the contribution of each party, which is made clear in our revised paper.
> > > > >
> > > > > >  I found the paper difficult to follow because there was no roadmap ... I do think much of my concerns can be alleviated if Section 3 was significantly clarified.
> > > > >
> > > > > We thank the reviewer for the feedback, and have included a **summarized description** of our mechanism in App. A to serve as the roadmap, due to the page limitations. We hope that our accordingly updated Sec. 3 provides better clarifications and has helped improve your opinion of our work.

---

### Author Response · Authors · 2023-11-21
**General Author Response**

## **Post-discussion period summary (of responses and updates)**

We sincerely appreciate the valuable feedback provided by the reviewers. We summarize the *latest* responses and updates to the manuscript:

1. We thoroughly **responded to all questions** from the reviewers to address their concerns. Both reviewers `1Wgz` and `gZai` asknowledge that their questions have been addressed.

    We note that Reviewers `1Wgz` and `hnks` primarily raised comments on the clarity of our presentation, which we try to address by providing clarifications in our response and updating the manuscript accordingly.

2. The following updates are included:
    - We *carefully revise* Sec. 3 (Reviewer `1Wgz`) to include Fig. 1 (Reviewer `gZai`) to *clearly demostrate* the components and steps of our proposed mechanism.
    - Further, we add a *detailed but concise* description of the precise steps in our mechanism in App. A (Reviewer `1Wgz`).

In the revision, we will continue to refine our writing and presentation to improve its clarity, by carefully incorporating the suggestions from the reviewers.

We wish to extend our sincere thanks to all the reviewers for their constructive feedback and for acknowledging that our response has addressed their questions. We hope that our clarifications and updates have addressed all the questions and helped raise your opinions of our work.


---
---

We are extremely grateful to the reviewers for their insightful comments, which are key to refining and improving our paper.
***
### **Strengths**

It is clear from the reviews that several aspects of our paper have been **positively accepted**:

- Our study on incentivized black-box model sharing is an **important**, **practical**, and **well-motivated** topic (Reviewers `1Wgz`, `gZai`, `hnks`).

- We have **clearly addressed** the three main questions of incentivized black-box model sharing (Reviewer `jHme`).

- We have provided empirical evaluations to **verify the theoretical claims** (Reviewers `gZai`, `hnks`).

- Our proposed properties are **desirable**, and the analysis is **thorough** (Reviewers `1Wgz`, `hnks`).

***
### **Revision**

We have made the following modifications to our paper in the rebuttal:

- We have conducted an additional experiment (update in App. D.1) and highlighted that the requirement on the ground truth is not very restrictive, where the labeled data (i.e., ground truth) of size as small as 100 is sufficient to identify high-quality models, as shown in Fig. 6 in App. D.1.

- We have provided additional empirical results in Fig. 8 in App. D.2.1 to demonstrate the improved accuracy with *different practical ensemble methods*, where the strict IR is still achieved.

- We have provided an empirical quantification between the **accuracy gain** w.r.t. the increase in (monetary) **payment** in Fig. 9 in App. D.2.1.


- We have updated our optimization problem of the optimal ensemble method in App. B.2 to achieve a similar weight $\beta_{i,x}$ for similar prediction $h_i(x)$.

- We have added a section (App. A.1) discussing the limitations.

- We have discussed two more properties of Shapley value (App. B.1).

***
We will carefully consider all the feedback given in our revised version. We thank the reviewers for their valuable input and hope our answers can increase your opinions of our work. We are happy to provide more clarifications.

---

### Meta-Review · Area_Chair_qVBs · 2023-12-12

**Metareview:**

Reviewers appreciated the extensive effort by the authors to clarify their contributions, but they also maintained several concerns regarding the paper.

-The presentation clarity of the paper remains a major weakness, making it difficult to understand the overall model, and the applicability of the results. Expositional issues stem from the fact that the model / framework is very complex. The complexity is also an issue in itself, as it is hard to pinpoint exactly where the contribution of the paper lies, what assumptions are being made, and why they are justified.
-On the technical side, the fact that getting a good set of weights is hard should at the very least be discussed.
-Most importantly however, as the quality of the guarantees depends on the weights, and the weights are purportedly learned from a small dataset, there should be some discussion of the distribution from which it is sampled (especially given that the introduction mentions agents covering different distributions), and how bounds depend on sampling from this distribution. Some argument should also be made as to why predictions (and ensemble weights) generalize beyond this small dataset.

For all this reasons, the paper could use another iteration prior to publication.

**Justification For Why Not Higher Score:**

Reviewers were very explicit about their concerns during the discussion.

**Justification For Why Not Lower Score:**

N/A

---

### Decision · Program_Chairs · 2024-01-16

Reject